# Saving Foundation Flow-Matching Priors for Inverse Problems

**Yuxiang Wan** [1]  **Ryan Devera** [1]  **Wenjie Zhang** [1]  **Ju Sun** [1]

Project page: `https://sun-umn.github.io/xm-plug/`

## Abstract

Foundation flow-matching (FM) models promise universal priors for solving inverse problems (IPs); yet today, they trail behind domain-specific and even untrained priors. *How can we unlock their potential?* We introduce FMPlug, a plug-in framework that redefines how foundation FMs are used in IPs. FMPlug combines an instance-guided, time-dependent warm-start strategy with sharp Gaussianity regularization, adding problem-specific guidance while preserving the Gaussian structures. For evaluation, we consider both simple image restoration tasks and scientific IPs with a few similar samples—where the prohibitive cost of data collection and model training hinders the development of domain-specific generative models. Our superior experimental results confirm the effectiveness of FMPlug. Overall, FMPlug paves the way for making foundation FM models practical, reusable priors for IPs, especially scientific ones with few similar samples.

## 1. Introduction

Inverse problems (IPs) are prevalent in many fields, such as medical imaging, remote sensing, and computer vision (Aster et al., 2018; Mohamad-Djafari, 2013). In an IP, the objective is to recover an unknown object $\boldsymbol{x}$ of interest from the relevant measurement $\boldsymbol{y} \approx \mathcal{A}(\boldsymbol{x})$, where the mapping $\mathcal{A}(\cdot)$, called the **forward model**, represents the measurement process, and the approximation sign $\approx$ accounts for possible modeling errors and measurement noise. Due to insufficient measurement and/or the approximate relationship in $\boldsymbol{y} \approx \mathcal{A}(\boldsymbol{x})$, in practice, $\boldsymbol{x}$ is typically not uniquely recoverable from $\boldsymbol{y}$ alone, i.e., it is ill-posed. To obtain reliable, meaningful solutions for IPs, it is important to incorporate

[1]Department of Computer Science and Engineering, University of Minnesota, Minneapolis, Minnesota, USA. Correspondence to: Yuxiang Wan <wan01530@umn.edu>, Ju Sun <jusun@umn.edu>.

*Proceedings of the 43rd International Conference on Machine Learning*, Seoul, South Korea. PMLR 306, 2026. Copyright 2026 by the author(s).

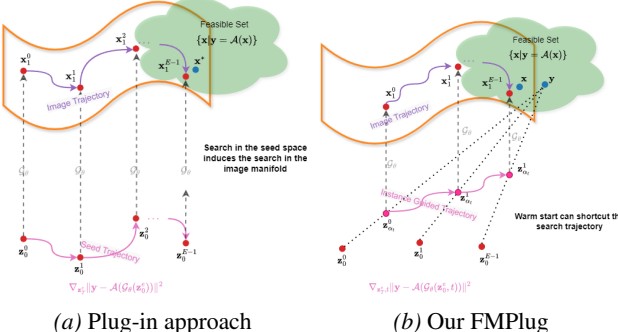

*(a)* Plug-in approach          *(b)* Our FMPlug

*Figure 1.* Illustration of the general plug-in approach (left) and our FMPlug method (right) for solving IPs with pretrained foundation FM priors.

prior knowledge of $\boldsymbol{x}$.

Traditional ideas for solving IPs rely on optimization, often motivated by the Maximum A Posteriori (MAP) principle:

$$\min_{\boldsymbol{x}}\ \ell(\boldsymbol{y}, \mathcal{A}(\boldsymbol{x})) + \Omega(\boldsymbol{x}). \tag{1.1}$$

Here, minimizing the data fitting loss $\ell(\boldsymbol{y}, \mathcal{A}(\boldsymbol{x}))$ encourages $\boldsymbol{y} \approx \mathcal{A}(\boldsymbol{x})$, and the regularization term $\Omega(\boldsymbol{x})$ encodes prior knowledge of ideal solutions to resolve ambiguities and hence mitigate potential ill-posedness. The resulting optimization problems are often solved by gradient-based iterative methods. **Advances in deep learning (DL) have revolutionized IP solving**. Different DL-based approaches to IPs involve varying levels of data-knowledge trade-offs. For example, supervised approaches take paired datasets $\{(\boldsymbol{y}_i, \boldsymbol{x}_i)\}_{i=1,\ldots,N}$ and directly learn the inverse mapping $\boldsymbol{y} \mapsto \boldsymbol{x}$, with or without using $\mathcal{A}$ (Ongie et al., 2020; Monga et al., 2021; Zhang et al., 2024); alternatively, data-driven priors learned from object-only datasets $\{\boldsymbol{x}_i\}_{i=1,\ldots,N}$ can be integrated with Eq. (1.1) to form hybrid optimization formulations that effectively combine data-driven priors on $\boldsymbol{x}$ and knowledge about $\mathcal{A}$, noise, and other aspects (Oliviero-Durmus et al., 2025; Daras et al., 2024; Wang et al., 2024; 2025); strikingly, untrained DL models themselves can serve as effective plug-in priors for Eq. (1.1), without any extra data (Alkhouri et al., 2024; 2025; Wang et al., 2023; Li et al., 2023; Zhuang et al., 2023a;b; Li et al., 2021).(Ongie et al., 2020; Monga et al., 2021; Alkhouri et al., 2025; Scarlett et al., 2023; Daras et al., 2024; Oliviero-Durmus et al., 2025; Vyas et al., 2024; Liang et al., 2025) give comprehensive

reviews of these DL-based ideas.

In this paper, **we focus on solving IPs with deep generative priors (DGPs) pretrained on object-only datasets** (Oliviero-Durmus et al., 2025). Compared to supervised approaches that require constructing task-specific paired datasets and performing task-specific training, this approach offers greater flexibility, as DGPs can be plugged into and reused for different IP problems within the same family of objects. Among the different DGPs, **we are most interested in those based on the emerging flow-matching (FM) framework (Lipman et al., 2024)**—which is rapidly superseding diffusion models as the backbone of state-of-the-art (SOTA) deep generative models (Black Forest Labs et al, 2025; Patrick Esser et al, 2024; Agarwal, Niket et al, 2025) due to its simple concepts and superior performance.

Several recent works have proposed solving IPs using pretrained FM priors (Daras et al., 2024). Although promising, most of them are based on **domain-specific** FM priors, e.g., trained on the FFHQ dataset for human faces and the LSUN bedrooms dataset for bedroom scenes. This limits the practicality of these methods, as domain-specific FM models are not always readily available, e.g., due to data or computing constraints. On the other hand, the emergence of domain-agnostic **foundation** FM models, such as the most recent Stable Diffusion and Flux models (Patrick Esser et al, 2024; Black Forest Labs et al, 2025) for images, obsoletes domain-specific developments. (Kim et al., 2025; Patel et al., 2024; Ben-Hamu et al., 2024; Martin et al., 2025) propose such ideas. **However, the performance reported from these works based on foundation FM priors clearly lags behind those with domain-specific FM priors, and even behind those with untrained priors**; see Sec. 2.3. This is not entirely surprising, as foundation priors are considerably weaker than domain-specific priors in terms of constraining the objects.

In this paper, we take the first step in closing the performance gap. We focus on IPs where the object $x$ is an image, as foundation FM models for images are widely available and image-related IPs find broad applications. To strengthen the foundation FM priors, we consider two practical settings: (A) **simple-distortion setting**, in which $x$ and $y$ are close (e.g., typical image restoration tasks); and (B) **few-shot setting**, in which a small number of image instances close to $x$ are provided (e.g., scientific IPs). For both settings, using the image instance(s) close to $x$ as a guide, we develop a time-dependent warm-start strategy and sharp Gaussian regularization, which together yield convincing performance gains. Our contributions include:

- **Identifying** the performance gap between foundation FM, domain-specific, and untrained priors for solving IPs (Sec. 2.3).
- **Proposing** a time-dependent warm-start strategy and a

sharp Gaussian regularization that effectively strengthen foundation FM priors (Sec. 3). We introduce a few-shot, instance-guided approach tailored for scientific IPs.
- **Confirming** the effectiveness of the proposed prior-strengthening method through systematic experimentation (Sec. 4). Our proposed method, **FMPlug**, achieves superior performance among pretrained foundation FM-based solvers for simple distortion and scientific IPs.

## 2. Technical background

### 2.1. Flow matching (FM)

Flow Matching (FM) models are an emerging class of deep generative models (Lipman et al., 2024). They learn a continuous flow to transform a prior distribution $p_0(z)$ into a target distribution $p_1(z)$—in the same spirit as continuous normalizing flow (CNF) (Chen et al., 2018; Grathwohl et al., 2019), where the flow is described by an ordinary differential equation (ODE)

$$d\boldsymbol{z} = \boldsymbol{v}(\boldsymbol{z}, t)\, dt. \qquad (2.1)$$

Whereas CNF focuses on the density path induced by the flow and performs maximum likelihood estimation as the learning objective, FM tries to learn a parameterized velocity field $\boldsymbol{v}_{\boldsymbol{\theta}}(\boldsymbol{z}, t)$ to match the one associated with the desired flow. To generate new samples after training, one simply samples $\boldsymbol{z}_0 \sim p_0(\boldsymbol{z})$ and numerically solves the learned ODE induced by $\boldsymbol{v}_{\boldsymbol{\theta}}(\boldsymbol{z}, t)$ from $t = 0$ to $t = 1$, to produce a sample $\boldsymbol{z}_1 \sim p_1(\boldsymbol{z})$.

For tractability, in practice, FM matches the conditional velocity field instead of the unconditional one discussed above: for each training point $\boldsymbol{x}$, a simple conditional probability path $p_t(\boldsymbol{z}_t|\boldsymbol{x})$, e.g., induced by a linear flow $\boldsymbol{z}_t = t\boldsymbol{x} + (1-t)\boldsymbol{z}_0$, is defined. The model $\boldsymbol{v}_{\boldsymbol{\theta}}(\boldsymbol{z}_t, t)$ is then trained to learn the known vector field of these conditional flows, i.e., $\boldsymbol{u}(\boldsymbol{z}_t, t|\boldsymbol{x})$:

$$\min_{\boldsymbol{\theta}}\ \mathbb{E}_{\boldsymbol{x}, \boldsymbol{z}_0, t} \left\| \boldsymbol{v}_{\boldsymbol{\theta}}(\boldsymbol{z}_t, t) - \boldsymbol{u}(\boldsymbol{z}_t, t|\boldsymbol{x}) \right\|^2. \qquad (2.2)$$

Diffusion models based on probability flow ODEs can also be interpreted as FMs, although (1) they match the score functions $\nabla_{\boldsymbol{z}} \log p_t(\boldsymbol{z})$ induced by the chosen probability path, not the velocity field as in FM; and (2) they typically work with affine flows for convenience, instead of the simple linear flows often used in FM practice (Lipman et al., 2024; Song et al., 2021). So, FM can be viewed as a general deep generative framework that covers diffusion models.

### 2.2. Pretrained FM priors for IPs

Recent methods that use pretrained FM priors for solving IPs can be classified into two families: **(1) The interleaving approach** interleaves the ODE generation steps

(i.e., numerical integration steps) with gradient steps toward measurement feasibility (i.e., moving $x$ around to satisfy $y \approx \mathcal{A}(x)$) (Pokle et al., 2023; Kim et al., 2025; Patel et al., 2024; Martin et al., 2025; Erbach et al., 2025). Despite the simplicity and empirical effectiveness on simple IPs, these methods might not converge or return an $x$ that respects the pretrained FM prior (i.e., **manifold feasibility**) or satisfies the measurement constraint $y \approx \mathcal{A}(x)$ (i.e., **measurement feasibility**); and **(2) The plug-in approach** views the generation process as a function $\mathcal{G}_\theta$ that maps any source sample to a target sample, and plugs the prior into Eq. (1.1) to obtain a unified formulation (Ben-Hamu et al., 2024):

$$z^* \in \arg\min_z \; \ell(y, \mathcal{A} \circ \mathcal{G}_\theta(z)) + \Omega \circ \mathcal{G}_\theta(z), \quad (2.3)$$

where $\circ$ denotes functional composition. The estimated object is $\mathcal{G}_\theta(z^*)$. Here, the generator $\mathcal{G}_\theta$ is fixed, and the output $\mathcal{G}_\theta(z)$ naturally satisfies manifold feasibility. In addition, global optimization of Eq. (2.3) forces small $\ell(y, \mathcal{A} \circ \mathcal{G}_\theta(z))$, and hence $y \approx \mathcal{A} \circ \mathcal{G}_\theta(z)$, i.e., leading to measurement feasibility. We note that there is a similar classification of recent work using pretrained diffusion priors to solve IPs; see (Wang et al., 2024; 2025; Daras et al., 2024; Oliviero-Durmus et al., 2025).

### 2.3. Foundation FM priors for IPs

*Table 1.* Comparison between foundation FM, domain-specific FM, and untrained priors for Gaussian deblurring the on AFHQ-Cat (resolution: $256 \times 256$). DS: domain-specific FM; FD: foundation FM; FD-S: strengthened foundation FM; DIP: deep image prior. **Bold**: best, & underline: second best, for each metric/column. The foundation model is Stable Diffusion V3 here.

| | PSNR↑ | SSIM↑ | LPIPS↓ | CLIPIQA↑ |
|---|---|---|---|---|
| **DIP** | 27.5854 | 0.7179 | 0.3898 | 0.2396 |
| **D-Flow (DS)** | **28.1389** | **0.7628** | **0.2783** | **0.5871** |
| **D-Flow (FD)** | 25.0120 | 0.7084 | 0.5335 | 0.3607 |
| **D-Flow (FD-S)** | 25.1453 | 0.6829 | 0.5213 | 0.3228 |
| **FlowDPS (DS)** | 22.1191 | 0.5603 | 0.3850 | 0.5417 |
| **FlowDPS (FD)** | 22.1404 | 0.5930 | 0.5412 | 0.2906 |
| **FlowDPS (FD-S)** | 22.0538 | 0.5920 | 0.5408 | 0.2913 |

#### 2.3.1. FOUNDATION FM PRIORS ≪ DOMAIN-SPECIFIC AND EVEN UNTRAINED ONES

The availability of large-scale training sets has recently fueled the intensive development of foundation generative models in several domains, most of them based on FM models and variants, e.g., Stable Diffusion V3 (and newer) (Patrick Esser et al, 2024) and FLUX.1 (Black Forest Labs et al, 2025) for images, OpenAI Sora (OpenAI, 2024) and Google Veo (DeepMind, 2025) for videos, and Nvidia Cosmos world model (Agarwal, Niket et al, 2025). By contrast, domain-specific FM models are not always

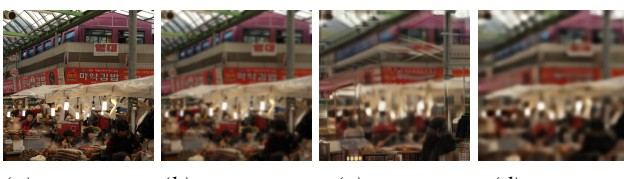

*(a)* Ground Truth   *(b)* Measurement   *(c)* FlowDPS   *(d)* FlowChef

*Figure 2.* Illustration of foundation FM prior used for Gaussian deblurring on DIV2k (resolution: $512 \times 512$). The reconstruction quality by FlowDPS and FlowChef is no better than that of the blurry image, i.e., measurement, itself, if not worse.

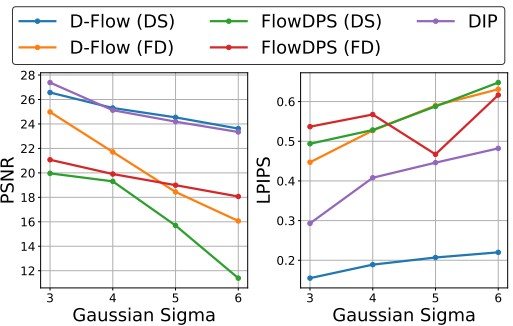

*Figure 3.* Comparison between foundation FM, domain-specific FM, and untrained priors for Gaussian deblurring with varying kernel size (Gaussian sigma) and hence varying difficulty level. Notations are the same as in Tab. 1.

readily available (e.g., due to the lack of training data for scientific applications). Recent IP methods based on pretrained FM priors have begun to explore foundational priors.

Although recent foundation FM models are powerful enough to generate diverse objects, when used as object priors for IPs, they only constrain the object to be physically meaningful (e.g., the object being a natural image)—**foundation models are powerful as generative models, but not specific as generative priors to impose informative constraints**. In comparison, domain-specific priors provide much more semantic and structural information about the object (e.g., that it is a facial or brain MRI image). Thus, **foundation priors alone are considerably weaker than domain-specific priors for IPs**. In fact, untrained priors, such as vanilla deep image prior (DIP) and implicit neural representation, may be powerful enough to promote physically meaningful solutions for IPs (Alkhouri et al., 2025; Wang et al., 2023; Li et al., 2023; Zhuang et al., 2023a;b; Sitzmann et al., 2020).

A quick comparison summarized in Tab. 1 confirms our intuition: **recent IP methods with foundation FM priors perform much worse than domain-specific FM and even untrained priors** on Gaussian deblurring. Here, Flow-DPS (Kim et al., 2025) and D-Flow (Ben-Hamu et al., 2024) are representative interleaving and plug-in IP methods, respectively. For both of them, foundation priors (`FlowDPS(FD)` & `D-Flow(FD)`) lag behind domain-specific (`FlowDPS(DS)` & `D-Flow(DS)`) priors by considerable mar-

gins in at least two of the four metrics (see Fig. 2 for visualization of failures). Moreover, Eq. (1.1) integrated with the untrained DIP is the second best method by three of the four metrics, just after `D-Flow(DS)`. Similarly, results on Gaussian deblurring with varying kernel size presented in Fig. 3 show unequivocally that domain-specific FM and untrained priors are stronger than foundation FM priors, uniformly across different difficulty levels of Gaussian deblurring.

### 2.3.2. CURRENT IDEAS TO STRENGTH FOUNDATION FM PRIORS DO NOT QUITE WORK

Although none of the previous work **explicitly** acknowledges and discusses the serious performance issue of foundation FM priors, some have **implicitly** tried to strengthen the priors. As a plug-in method, (Ben-Hamu et al., 2024) assumes that $x$ and $y$ are close—e.g., valid for typical image restoration tasks, and initializes the optimization variable $z$ of Eq. (2.3) with

$$z_0 = \sqrt{\alpha} y_0 + \sqrt{1-\alpha} z \quad \text{with } z \sim \mathcal{N}(\mathbf{0}, \boldsymbol{I}), \quad (2.4)$$

where $y_0$ is the **inversion seed**, i.e., $y_0 = y + \int_1^0 v_\theta(y_t, t) dt$—backward solution of the governing ODE, **to accelerate the convergence of numerical methods for solving Eq. (2.3)**. Moreover, they promote the Gaussianity of the seed $z_0$ by recognizing that $\|z_0\|_2^2$ follows a $\chi^2$ distribution and thus regularizing its negative log-likelihood. Alternatively, as a representative interleaving method, (Kim et al., 2025) also assumes the closeness of $x$ and $y$, and takes an automatically generated text description for $y$ as the text condition for the FM prior, as all recent foundation FM models allow text-prompted generation. However, **our quick empirical evaluation suggests that these prior-strengthening techniques are almost useless**: there is little change in performance moving from `FlowDPS(FD)` & `D-Flow(FD)` to `FlowDPS(FD-S)` & `D-Flow(FD-S)` in Tab. 1.

## 3. Method

The goal of this paper is to close the performance gap between foundation FM priors and domain-specific FM & untrained ones identified in Sec. 2.3.1, by addressing the deficiencies of the current prior-strengthening ideas summarized in Sec. 2.3.2. Between the two approaches to solving IPs with pretrained FM priors (Sec. 2.2), we follow the **plug-in approach** as formulated in Eq. (2.3), due to its superior performance in practice (see, e.g., Tab. 1 and Sec. 4). For this approach, a potential concern is whether $\mathcal{G}_\theta$ is surjective, i.e., whether every reasonable $x$ can be represented as $\mathcal{G}_\theta(z)$ for a cer-

*Table 2.* Image regression on 1000 random images from the `DIV2K` dataset; details in App. A.2.

| Metric | D-Flow | FMPlug |
|--------|--------|--------|
| PSNR   | 36.187 | 37.924 |
| LPIPS  | 0.181  | 0.093  |

tain $z$. While theoretical results of this nature seem lacking and modeling high-dimensional distributions for such theoretical analysis also seems tricky, empirically, the desired surjectivity seems to hold approximately based on our image regression test reported in Tab. 2.

To strengthen the foundation FM priors, we consider two practical settings: (A) the **simple-distortion setting**, in which $x$ and $y$ are close, e.g., for image restoration. This is the setting considered in previous prior-strengthening works (Ben-Hamu et al., 2024; Kim et al., 2025); and (B) the **few-shot setting**, in which a small number of image instances close to $x$ are provided, but $x$ and $y$ might not be close. This is particularly relevant for IPs arising from scientific imaging, where the image domain is typically very narrow and known with a few samples (Huang et al., 2022; Shen et al., 2019; Masto et al., 2025). For both settings, taking the image instance(s) close to $x$ as a guide, we develop a time-dependent warm-start strategy and a sharp Gaussian regularization that together lead to convincing performance gains. Below, we first describe the warm-start strategy and the Gaussian regularization for the simple-distortion setting in Sec. 3.1 and Sec. 3.2, respectively; we then discuss how to extend the ideas to deal with the few-shot setting in Sec. 3.4.

Gaussianity in the source and intermediate distributions of FM models, especially the following celebrated concentration-of-measure (CoM) result for Gaussian vectors, is crucial for our method.

**Theorem 3.1** (Concentration of measure in Gaussian random vectors (Vershynin, 2018)). *For a $d$-dimensional Gaussian random vector $z \sim \mathcal{N}(\mathbf{0}, \boldsymbol{I}_d)$, $\mathbb{P}[|\|z\|_2 - \sqrt{d}| \geq t] \leq 2e^{-ct^2}$ for a universal constant $c > 0$.*

This implies that for a standard Gaussian vector $z \in \mathbb{R}^d$, $\|z\|_2$ lies sharply in the range $[(1-\varepsilon)\sqrt{d}, (1+\varepsilon)\sqrt{d}]$ with $\varepsilon \in o(1)$ with overwhelmingly high probability. In other words, $z$ lies in an ultra-thin shell around $\mathbb{S}^{d-1}(\mathbf{0}, \sqrt{d})$ (a sphere in $\mathbb{R}^d$ centered at $\mathbf{0}$ and with a radius $\sqrt{d}$).

### 3.1. Instance-guided & time-dependent warm-start

**Why is the initialization strategy in D-Flow problematic?** In the standard FM setting, the source distribution $z_0 \sim \mathcal{N}(\mathbf{0}, \boldsymbol{I})$, whereas the initialized $z_0$ in Eq. (2.4) has a distribution $\mathcal{N}(\sqrt{\alpha} y_0, (1-\alpha)\boldsymbol{I})$. One might not worry about this distribution mismatch, as both are supported on the entire ambient space in theory. But finite-sample training with polynomially many samples in practice causes a significant gap: due to CoM of Gaussian vectors (Thm. 3.1), virtually all training samples drawn from $\mathcal{N}(\mathbf{0}, \boldsymbol{I})$ come from an ultra-thin shell $\mathcal{S}$ around $\mathbb{S}^{d-1}(\mathbf{0}, \sqrt{d})$,[1] so the generation function $\mathcal{G}_\theta$ is effectively

---

[1]To be precise, for $m$ iid drawn Gaussian vectors $z_1, \ldots, z_m$, $\mathbb{P}[\exists i \in \{1, \ldots, m\}$ with $|\|z_i\|_2 - \sqrt{d}| \geq t] \leq 2me^{-ct^2} = $

trained on inputs from the domain $\mathcal{S}$, not the entire ambient space—which implies that the behavior of $\mathcal{G}_{\boldsymbol{\theta}}$ on $\mathcal{S}^c$, the complement of $\mathcal{S}$, is largely undetermined. Now, samples from $\mathcal{N}(\sqrt{\alpha}\boldsymbol{y}_0, (1-\alpha)\boldsymbol{I})$ concentrate around another ultra-thin shell around $\mathbb{S}^{d-1}(\sqrt{\alpha}\boldsymbol{y}_0, \sqrt{(1-\alpha)d})$, which has only a negligibly small intersection with $\mathcal{S}$ and lies mostly in $\mathcal{S}^c$. So, the initialization in Eq. (2.4) lies in $\mathcal{S}^c$ with very high probability. Given that the behavior of $\mathcal{G}_{\boldsymbol{\theta}}$ on $\mathcal{S}^c$ can be wild, this initialization strategy is problematic.

**Our time-dependent warm-up strategy** A typical flow of FM models takes the form

$$\boldsymbol{z}_t = \alpha_t \boldsymbol{x} + \beta_t \boldsymbol{z} \quad \text{where } \boldsymbol{z} \sim \mathcal{N}(\boldsymbol{0}, \boldsymbol{I}), \qquad (3.1)$$

where $\alpha_t$ and $\beta_t$ are known functions of $t$ with the property

$$\alpha_t \searrow 0, \beta_t \nearrow 1 \text{ as } t \to 0, \text{ and } \alpha_t \nearrow 1, \beta_t \searrow 0 \text{ as } t \to 1, \quad (3.2)$$

where $\nearrow$ and $\searrow$ indicate monotonically increasing and decreasing, respectively. When $\boldsymbol{x}$ and $\boldsymbol{y}$ are close, $\boldsymbol{x} = \boldsymbol{y} + \boldsymbol{\varepsilon}$ for some small (i.e., $\|\boldsymbol{\varepsilon}\|$ is small compared to $\|\boldsymbol{x}\|$ and $\|\boldsymbol{z}\|$) but unknown $\boldsymbol{\varepsilon}$. So, we can write the flow as

$$\boldsymbol{z}_t = \alpha_t(\boldsymbol{y} + \boldsymbol{\varepsilon}) + \beta_t \boldsymbol{z} = \alpha_t \boldsymbol{y} + \beta_t \boldsymbol{z} + \alpha_t \boldsymbol{\varepsilon} \qquad (3.3)$$

where $\boldsymbol{z} \sim \mathcal{N}(\boldsymbol{0}, \boldsymbol{I})$. To eliminate the unknown $\boldsymbol{\varepsilon}$, we can approximate the exact flow in Eq. (3.3) by

$$\boldsymbol{z}_t \approx \alpha_t \boldsymbol{y} + \beta_t \boldsymbol{z} \quad \text{where } \boldsymbol{z} \sim \mathcal{N}(\boldsymbol{0}, \boldsymbol{I}) \qquad (3.4)$$

with an approximation error $\alpha_t \boldsymbol{\varepsilon}$. To control the error, (1) if $\boldsymbol{\varepsilon}$ is relatively large, a small $\alpha_t$ is desirable; (2) if $\boldsymbol{\varepsilon}$ is already relatively small, a relatively large $\alpha_t$ is acceptable. So, although we do not know $\boldsymbol{\varepsilon}$ itself and hence its magnitude, with appropriate $\alpha_t$ we can always make $\alpha_t \boldsymbol{\varepsilon}$ sufficiently small. So, we leave $\alpha_t$ as a learnable parameter. Since $\alpha_t$ is a known function of $t$, we simply need to leave $t \in [0, 1]$ learnable, leading to our warm-start formulation

$$\min_{\boldsymbol{z}, t \in [0,1]} \ell(\boldsymbol{y}, \mathcal{A} \circ \mathcal{G}_{\boldsymbol{\theta}}(\alpha_t \boldsymbol{y} + \beta_t \boldsymbol{z}, t)). \qquad (3.5)$$

Here we overload the notation of $\mathcal{G}_{\boldsymbol{\theta}}$ as $\mathcal{G}_{\boldsymbol{\theta}} : \mathbb{R}^d \times [0,1] \to \mathbb{R}^d$—the second input is the current $t$ on the path (the notation in Eq. (2.3) assumes $t = 0$). In other words, due to the closeness of $\boldsymbol{x}$ and $\boldsymbol{y}$, we do not need to start from scratch, i.e., $t = 0$; instead, we plug $\boldsymbol{y}$ into an appropriate, learnable time point to create a shortcut, as illustrated in Fig. 1a. Our warm-start strategy is not only grounded in theory and effective in practice (see Sec. 4), but also speeds up learning, as $t > 0$ implies shorter flows.

### 3.2. Sharp Gaussianity regularization

**Why is the Gaussian regularization in D-Flow problematic?** If $\boldsymbol{z}_0 \sim \mathcal{N}(\boldsymbol{0}, \boldsymbol{I})$, $\|\boldsymbol{z}_0\|_2^2 \sim \chi^2(d)$ and the neg-

---

$2e^{-ct^2 + \log m}$ by a simple union bound. Taking $t = \varepsilon \sqrt{d}$, we obtain that $\mathbb{P}[\exists i \in \{1, \dots, m\} \text{ with } |\|\boldsymbol{z}_i\|_2 - \sqrt{d}| \geq \varepsilon\sqrt{d}] \leq 2e^{-c\varepsilon^2 d + \log m} \leq 2e^{-c\varepsilon^2 d/2}$ provided that $m \leq e^{c\varepsilon^2 d/2}$.

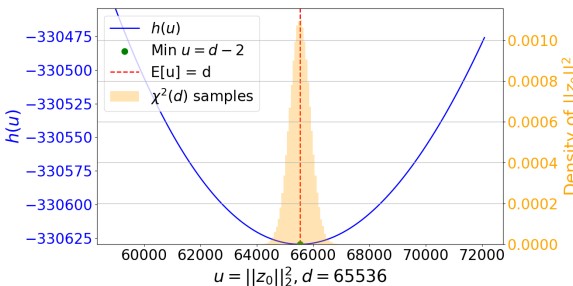

*Figure 4.* Plot of the function $h(\boldsymbol{z}_0)$ (after a change of variable $u = \|\boldsymbol{z}_0\|_2^2$). An ideal regularization function should blow up sharply away from the narrow concentration region in orange to promote Gaussianity effectively.

ative log-likelihood is $h(\boldsymbol{z}_0) = -(d/2 - 1) \log \|\boldsymbol{z}_0\|_2^2 + \|\boldsymbol{z}_0\|_2^2 / 2 + C$ for some constant $C$ independent of $\boldsymbol{z}_0$. (Ben-Hamu et al., 2024) promotes the Gaussianity of $\boldsymbol{z}_0$ by regularizing $h(\boldsymbol{z}_0)$. While $h(\boldsymbol{z}_0)$ is minimized at any $\boldsymbol{z}_0$ satisfies $\|\boldsymbol{z}_0\|_2 = \sqrt{d - 2}$, away from this value the function changes painfully slowly; see Fig. 4. For example, the function value only changes $\leq 0.031\%$ relative to the minimum in the $[62000, 70000]$ range, much larger than the orange-highlighted CoM region. This is problematic, as $\|\boldsymbol{z}_0\|_2$ should concentrate sharply around $d$ and thus only functions that blow up quickly away from the $\|\boldsymbol{z}_0\|_2 = \sqrt{d}$ level can effectively promote the Gaussianity of $\boldsymbol{z}_0$.

**Our sharp Gaussian regularization via an explicit constraint** For Eq. (3.5), we hope to promote the Gaussianity of $\boldsymbol{z}$. To enforce the sharp concentration of $\boldsymbol{z}$, we introduce the shell constraint

$$\mathbb{S}_{\varepsilon}^{d-1}(\boldsymbol{0}, \sqrt{d}) \doteq \{\boldsymbol{z} : \|\boldsymbol{z}\|_2 \in [1 - \varepsilon, 1 + \varepsilon]\sqrt{d}\}, \quad (3.6)$$

with an $\varepsilon \ll 1$, as implied by Thm. 3.1. So, our final formulation for the simple-distortion setting is

$$\boxed{\begin{aligned} &\min_{\boldsymbol{z}, t \in [0,1]} \ \ell\left(\boldsymbol{y}, \mathcal{A} \circ \mathcal{G}_{\boldsymbol{\theta}}\left(\alpha_t \boldsymbol{y} + \beta_t \boldsymbol{z}, t\right)\right) \\ &\text{s.t. } \boldsymbol{z} \in \mathbb{S}_{\epsilon}^{d-1}(\boldsymbol{0}, \sqrt{d}) \end{aligned}} \quad (3.7)$$

Note that this still falls under the plug-in framework laid out in Eq. (2.3), as the constraint $\boldsymbol{z} \in \mathbb{S}_{\epsilon}^{d-1}(\boldsymbol{0}, \sqrt{d})$ can be equivalently written as a regularization term on $\boldsymbol{z}$ via the set-indicator function

$$\delta_{\mathbb{S}_{\epsilon}^{d-1}(\boldsymbol{0}, \sqrt{d})}(\boldsymbol{z}) = \begin{cases} 0 & \boldsymbol{z} \in \mathbb{S}_{\epsilon}^{d-1}(\boldsymbol{0}, \sqrt{d}) \\ \infty & \boldsymbol{z} \notin \mathbb{S}_{\epsilon}^{d-1}(\boldsymbol{0}, \sqrt{d}) \end{cases}. \quad (3.8)$$

To ensure feasibility, in each iteration step to optimize Eq. (3.7), we simply need to add the closed-form projection

$$\begin{aligned} &\mathcal{P}_{\mathbb{S}_{\epsilon}^{d-1}(\boldsymbol{0}, \sqrt{d})}(\boldsymbol{z}) \\ &= \begin{cases} (1 + \epsilon)\sqrt{d} \cdot \frac{\boldsymbol{z}}{\|\boldsymbol{z}\|_2} & \text{if } \|\boldsymbol{z}\|_2 \geq (1 + \epsilon)\sqrt{d} \\ (1 - \epsilon)\sqrt{d} \cdot \frac{\boldsymbol{z}}{\|\boldsymbol{z}\|_2} & \text{if } \|\boldsymbol{z}\|_2 \leq (1 - \epsilon)\sqrt{d} , \\ \boldsymbol{z} & \text{otherwise} \end{cases} \end{aligned} \quad (3.9)$$

where $\mathcal{P}_S(\cdot)$ denotes the Euclidean projection operator onto a set $S$. Using a spherical constraint $\|z\|_2 = \sqrt{d}$ or regularization to promote Gaussianity is not new in the FM and diffusion literature; see, e.g., Yang et al. (2024). However, enforcing $\|z\|_2 = \sqrt{d}$ is a bit rigid, as the actual length lies in a small range around it. Our shell constraint leaves reasonable slackness while still sharply encoding the Gaussianity. We typically set $\varepsilon = 0.025$ in our implementation.

### 3.3. Implementation issues in FMPlug

**FMPlug with latent FM models** Our formulation in Eq. (3.7) can be easily generalized to latent FM models commonly used in practice. Assume that $\mathcal{E}$ and $\mathcal{D}$ are the pretrained encoder and decoder, respectively, for the latent model. We just need to replace $\mathcal{A} \circ \mathcal{G}_{\boldsymbol{\theta}}$ in Eq. (3.7) with $\mathcal{A} \circ \mathcal{D} \circ \mathcal{G}_{\boldsymbol{\theta}}$, and $\boldsymbol{y}$ inside the warm-start term of Eq. (3.7) with $\mathcal{E}(\boldsymbol{y})$, yielding:

$$\min_{\boldsymbol{z},t\in[0,1]} \ell\left(\boldsymbol{y}, \mathcal{A} \circ \mathcal{D} \circ \mathcal{G}_\theta\left(\alpha_t \mathcal{E}(\boldsymbol{y}) + \beta_t \boldsymbol{z}, t\right)\right) \\ \text{s.t. } \boldsymbol{z} \in \mathbb{S}_\epsilon^{d-1}(\boldsymbol{0}, \sqrt{d}) \quad . \quad (3.10)$$

**Dimension mismatch in $x$ and $y$** In the simple-distortion setting, our assumption is that the target signal $\boldsymbol{x}$ and the observation $\boldsymbol{y}$, or **a tractable transformation of $\boldsymbol{y}$**, are close. Consequently, whenever feasible, we apply a deterministic transformation to $\boldsymbol{y}$ so that its dimension matches that of $\dim(\boldsymbol{x})$. For example, in super-resolution tasks, $\boldsymbol{y}$ is first upsampled to the target resolution. More broadly, for compressed sensing and other linear inverse problems where $\dim(\boldsymbol{y}) \neq \dim(\boldsymbol{x})$, we use $\boldsymbol{A}^\dagger \boldsymbol{y}$, where $\boldsymbol{A}^\dagger$ denotes the Moore-Penrose pseudoinverse of the known forward operator $\boldsymbol{A}$. In scenarios where defining such a simple transformation is intractable, the core simple-distortion assumption is generally violated; one could instead consider the following few-shot setting.

### 3.4. Few-shot setting for scientific IPs

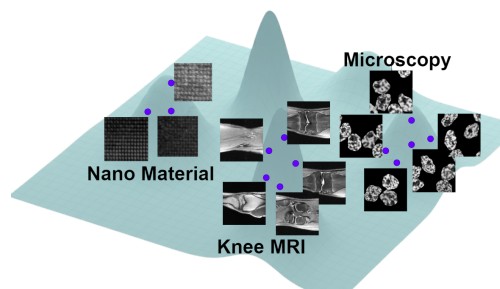

*Figure 5.* In scientific IPs, objects in the same domain are typically structurally similar and concentrated in a small region of the general image space.

**Why foundation FM priors for scientific imaging?** Many scientific domains rely on imaging for scientific discovery, e.g., materials science and astronomy, and often

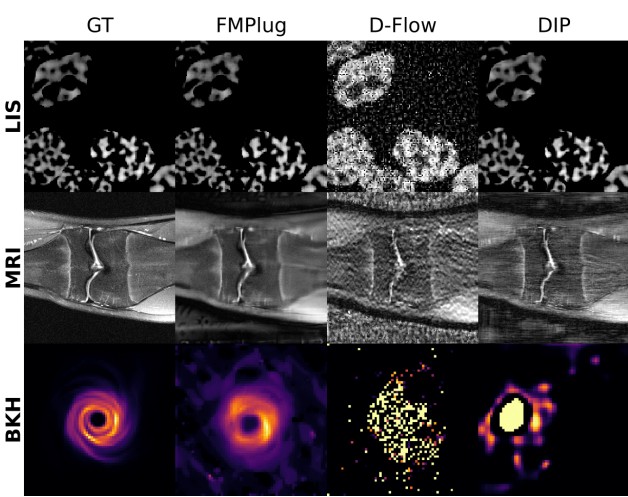

*Figure 6.* Three scientific IPs we focus on: LIS (linear inverse scatter- ing), MRI (compressed sensing MRI), and BKH (black hole imaging), and reconstruction results by three methods: our FMPlug, D-Flow, DIP (GT: groundtruth). More details in Sec. 4.2.

need to solve highly complex nonlinear IPs to interpret their measured data. Although it is tempting to train domain-specific FM models to help solve these IPs, there are practical barriers that are hard to overcome in the near term: (1) **prohibitive cost of data collection**. This could be due to difficulty in obtaining clean groundtruth data (e.g., black holes, nano-materials), or the high price or long duration needed to obtain a sufficient amount of data for training FM models (e.g., numerous microscopy modalities (Chua et al., 2022)); (2) **prohibitive cost of model development and training**. Even if the data are sufficient, training FM models might be computationally intensive. For example, scientific applications often entail ultra-high-resolution images (e.g., MRI) or high dynamic range (e.g., remote sensing (Chen et al., 2025)), which invariably demand models of greater complexity than those in the natural-image domain. The increased complexity of data and models inevitably leads to high computational costs, intimidating domain scientists (Bommasani et al., 2022). However, scientific discovery should not be halted because such domain-specific models do not yet exist; the similarity of many scientific images to natural images—perhaps after appropriate transformations—suggests using available foundation FM models trained on web-scale images to solve IPs in scientific imaging, as we demonstrate below.

**Our few-shot instance-guided strategy** We assume a small set of instances $\{\boldsymbol{x}_k\}_{k=1,\dots,K}$, some of which are close to the true $\boldsymbol{x}$—realistic for many scientific domains with limited visual variability, as illustrated in

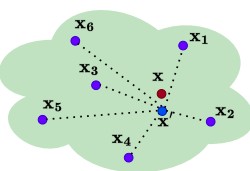

*Figure 7.* Illustration of few-shot instance prior.

Fig. 5. To adapt the time-dependent warm-start strategy in Sec. 3.1 to this setting, we consider linear combinations of $\boldsymbol{x}_k$'s to take the place of $\boldsymbol{y}$ for warm-start (see Fig. 7), i.e., starting with $\alpha_t(\sum_{k=1}^{K} w_k \boldsymbol{x}_k) + \beta_t \boldsymbol{z}$, resulting in

$$\boxed{\begin{aligned} &\min_{\boldsymbol{z},t,\boldsymbol{w}} \ \ell(\boldsymbol{y}, \mathcal{A} \circ \mathcal{G}_{\boldsymbol{\theta}}(\alpha_t \sum_{k=1}^{K} w_k \boldsymbol{x}_k + \beta_t \boldsymbol{z}, t)) \\ &\text{s.t. } \boldsymbol{z} \in \mathbb{S}_{\varepsilon}^{d-1}(\boldsymbol{0}, \sqrt{d}), t \in [0,1], \boldsymbol{w} \in \Delta^K \end{aligned}} \quad (3.11)$$

to replace Eq. (3.7). The simplex constraint $\boldsymbol{w} \in \Delta^K \doteq \left\{ \boldsymbol{w} \in \mathbb{R}^K : \boldsymbol{w} \geq \boldsymbol{0}, \boldsymbol{w}^{\mathsf{T}} \boldsymbol{1} = 1 \right\}$ serves two purposes: (1) It fixes the scale of $\boldsymbol{w}$, as the multiplicative relationship of $\alpha_t$ and $\boldsymbol{w}$ causes scale ambiguity—e.g., we can scale $\boldsymbol{w}$ by a factor $\xi$ and scale $\alpha_t$ by $1/\xi$ to obtain the same $\alpha_t(\sum_{k=1}^{K} w_k \boldsymbol{x}_k)$, and vise versa; (2) It encourages the model to concentrate weight on a subset of relevant data, effectively identifying and prioritizing instances that are structurally most similar to the unknown target $\boldsymbol{x}$. In practice, we eliminate this constraint by a simple reparameterization, $\boldsymbol{w} = \text{softmax}(\boldsymbol{v})$, and treat $\boldsymbol{v}$ as an optimization variable. Since the proposed modification in warm-start does not affect $\boldsymbol{z}$, our sharp Gaussian regularization in Sec. 3.2 can be directly integrated.

## 4. Experiments

In this section, we benchmark **FMPlug**'s performance on both simple-distortion (Sec. 4.1) and few-shot IPs (Sec. 4.2). In Sec. 4.3, we perform an ablation study to dissect the contributions of the two key algorithmic components.

### 4.1. Simple-distortion IPs

**Datasets, tasks, and evaluation metrics** We use 3 diverse datasets: DIV2K (Agustsson & Timofte, 2017), RealSR (Cai et al., 2019) and AFHQ (Choi et al., 2020), and take 100 random images out of each dataset. We set the image resolution to $512 \times 512$ by resizing and cropping the original. We consider **four linear IPs**: i) $4\times$ **super-resolution** from $128 \times 128$ to $512 \times 512$; ii) $70\%$ **random-mask inpainting**; iii) **Gaussian deblurring** with a kernel size of 61 and a standard deviation of 3.0; iv) **Motion deblurring** with a kernel size of 61 and an intensity of 0.5. We add Gaussian noise $\sigma = 0.03$ to all measurements. For evaluation metrics, we use four reference-based metrics, including PSNR for pixel-level difference, SSIM and DISTS for structural and textural similarity, LPIPS for perceptual difference, and two no-reference metrics, CLIPIQA & MUSIQ. The reference-based metrics are our primary metrics, and the no-reference metrics are secondary and just serve as tie-breakers. Results measured by DISTS and MUSIQ are included in App. A.4.

**Competing methods** We compare our FMPlug (**-W**: warm-start only, Number of Function Evaluations (NFE) = 3) with deep image prior (DIP) (Ulyanov et al., 2020)

(an untrained image prior), D-Flow (NFE = 6) (Ben-Hamu et al., 2024) (a SOTA plug-in method), FlowDPS (NFE = 28) (Kim et al., 2025) (a SOTA interleaving method), and FlowChef (NFE = 28) (Patel et al., 2024) (another SOTA interleaving method). For fairness, we use Stable Diffusion V3 (Patrick Esser et al, 2024) as the common backbone. Since FlowDPS and FlowChef integrate text prompts, we compare two variants with the prompts on and off, respectively; we use the postfix **-P** to indicate the prompt-enabled variants. We use the pretrained degradation-aware prompt extractor of (Wu et al., 2024b) to generate label-style text prompts (CFG scale = 2.0). Details on implementation and hyperparameters can be found in Sec. 3.3 and App. A.3, respectively.

**Results** Tab. 3 summarizes the quantitative results. We observe that: **(1)** Our FMPlug is the overall winner across all metrics except CLIPIQA, beating the untrained DIP—a strong baseline. FlowChef and FlowDPS, with text prompts, lag behind even the untrained DIP by large margins and generate visually blurry, oversmooth, and often hallucinated images, as shown in Fig. 12, highlighting the general struggle of interleaving methods to ensure simultaneous measurement and manifold feasibility; **(2)** For plug-in methods, our FMPlug improves upon D–Flow—our main competitor—by considerable margins based on all metrics except CLIPIQA, showing the solid advantage of our warm-start strategy and Gaussian regularization over theirs; and **(3)** FMPlug further improves PSNR and SSIM slightly over FMPlug-W, showing stronger visual quality. This confirms the benefits of the sharp Gaussianity regularization. More quantitative results, visualization, and discussion can be found in App. A.4.

### 4.2. Few-shot scientific IPs

**Experiment setup** Inversebench (Zheng et al., 2025) is specifically designed to expose the "scientific gap" that standard visual IPs fail to capture. It includes five distinct scientific IPs, each presenting a unique structural challenge. We consider three of them.[2] and take their data as necessary: **(1) Linear inverse scattering (LIS)**, a linear IP in optical microscopy, where the objective is to recover the unknown permittivity contrast $\boldsymbol{z} \in \mathbb{R}^n$ from measurements of the scattered light field $\boldsymbol{y}_{\text{sc}} \in \mathbb{C}^m$. We use 100 samples for evaluation and 10 samples as guidance; **(2) Compressed sensing MRI**, a linear IP to accelerate MRI scanning through subsampling. We use 94 samples for evaluation and 6 for guidance; **(3) Black hole imaging (BKH)**, a nonlinear IP that reconstructs black holes from measurements using Very Long Baseline Interferometry (VLBI) with non-additive

---

[2]We exclude two IPs from InverseBench: **full waveform inversion**, due to optimization instability caused by the forward operator—faced by their original paper also; and the **Navier-Stokes equation**, as it requires gradient-free optimization.

*Table 3.* Quantitative results on simple-distortion IPs. (**Bold**: best, under: second best, CLIP: CLIPIQA)

| task | method | AFHQ (512 × 512) | | | | DIV2K (512 × 512) | | | | RealSR (512 × 512) | | | |
|---|---|---|---|---|---|---|---|---|---|---|---|---|---|
| | | PSNR ↑ | SSIM ↑ | LPIPS ↓ | CLIP ↑ | PSNR ↑ | SSIM ↑ | LPIPS ↓ | CLIP ↑ | PSNR ↑ | SSIM ↑ | LPIPS ↓ | CLIP ↑ |
| 4× Super Resolution | DIP | 29.85 | 0.78 | 0.37 | 0.33 | 25.75 | 0.73 | 0.42 | 0.40 | **26.81** | 0.72 | 0.44 | 0.30 |
| | FlowChef-P | 29.23 | 0.79 | 0.38 | 0.64 | 25.08 | 0.71 | 0.43 | **0.60** | 25.89 | 0.71 | 0.43 | **0.44** |
| | FlowDPS-P | 28.75 | 0.76 | 0.37 | 0.37 | 24.92 | 0.69 | 0.42 | 0.51 | 26.11 | 0.71 | 0.43 | 0.34 |
| | D-Flow | 26.37 | 0.70 | 0.54 | 0.31 | 23.42 | 0.64 | 0.52 | 0.37 | 23.60 | 0.62 | 0.53 | 0.28 |
| | **FMPlug-W** | 30.13 | **0.81** | 0.34 | 0.18 | 25.77 | **0.74** | **0.38** | 0.24 | 26.58 | 0.73 | 0.39 | 0.16 |
| | **FMPlug** | **30.31** | **0.81** | **0.33** | 0.20 | **25.88** | **0.74** | **0.38** | 0.27 | 26.66 | **0.74** | **0.38** | 0.17 |
| Rand. Inpaint. 70% | DIP | **33.32** | **0.90** | **0.21** | 0.47 | 28.49 | **0.86** | 0.27 | 0.59 | 30.88 | **0.89** | 0.25 | 0.47 |
| | FlowChef-P | 29.27 | 0.77 | 0.41 | 0.57 | 24.67 | 0.67 | 0.46 | 0.50 | 25.81 | 0.69 | 0.45 | 0.35 |
| | FlowDPS-P | 27.63 | 0.73 | 0.41 | 0.43 | 24.01 | 0.65 | 0.47 | 0.54 | 25.68 | 0.69 | 0.47 | 0.36 |
| | D-Flow | 28.43 | 0.76 | 0.41 | 0.65 | 24.71 | 0.73 | 0.41 | 0.67 | 25.27 | 0.69 | 0.42 | **0.59** |
| | **FMPlug-W** | 32.75 | 0.88 | 0.37 | 0.63 | 28.82 | 0.85 | 0.33 | 0.68 | 31.30 | 0.88 | 0.28 | 0.56 |
| | **FMPlug** | 32.81 | 0.87 | 0.34 | **0.66** | **28.95** | 0.84 | 0.32 | **0.69** | **31.79** | **0.89** | 0.26 | 0.56 |
| Gaussian Deblur | DIP | 29.39 | 0.77 | **0.39** | 0.30 | 25.23 | 0.70 | 0.43 | **0.38** | 26.17 | 0.70 | 0.46 | 0.28 |
| | FlowChef-P | 23.84 | 0.63 | 0.54 | 0.28 | 20.41 | 0.49 | 0.62 | 0.23 | 21.42 | 0.51 | 0.63 | 0.19 |
| | FlowDPS-P | 24.15 | 0.60 | 0.49 | 0.23 | 20.23 | 0.46 | 0.58 | 0.32 | 21.21 | 0.47 | 0.59 | 0.22 |
| | D-Flow | 25.90 | 0.66 | 0.54 | **0.34** | 23.64 | 0.64 | 0.52 | 0.37 | 23.65 | 0.60 | 0.54 | **0.30** |
| | **FMPlug-W** | 30.38 | **0.79** | 0.40 | 0.22 | 26.05 | 0.72 | 0.43 | 0.29 | 27.05 | 0.72 | 0.44 | 0.21 |
| | **FMPlug** | **30.41** | **0.79** | **0.39** | 0.21 | **26.26** | **0.73** | **0.41** | 0.28 | **27.22** | **0.73** | **0.43** | 0.19 |
| Motion Deblur | DIP | 28.69 | 0.75 | 0.38 | 0.26 | 24.75 | 0.68 | 0.45 | 0.35 | 26.17 | 0.70 | 0.46 | 0.28 |
| | FlowChef-P | 24.77 | 0.66 | 0.50 | **0.37** | 21.27 | 0.54 | 0.57 | 0.34 | 22.50 | 0.56 | 0.56 | 0.26 |
| | FlowDPS-P | 24.81 | 0.64 | 0.46 | 0.28 | 21.07 | 0.51 | 0.54 | 0.39 | 22.50 | 0.55 | 0.55 | 0.27 |
| | D-Flow | 27.81 | 0.73 | 0.48 | 0.35 | 25.21 | 0.70 | 0.47 | **0.42** | 25.86 | 0.69 | 0.47 | **0.31** |
| | **FMPlug-W** | 30.10 | 0.79 | 0.39 | 0.26 | 26.83 | 0.74 | 0.40 | 0.36 | 28.01 | 0.76 | 0.40 | 0.28 |
| | **FMPlug** | **30.43** | **0.81** | **0.37** | 0.28 | **27.38** | **0.78** | **0.36** | **0.42** | **28.63** | **0.79** | **0.37** | 0.30 |

*Table 4.* (Scientific IPs) Performance on LIS, MRI and BKH. (**Bold**: best among non-DS priors; Background: with DS model)

| | LIS | | MRI | | | BKH | | |
|---|---|---|---|---|---|---|---|---|
| | PSNR↑ | SSIM↑ | PSNR↑ | SSIM↑ | Data misfit↓ | PSNR↑ | $\tilde{\chi}^2_{logca}$ ↓ | $\tilde{\chi}^2_{cp}$ ↓ |
| DIP | 33.45 | 0.90 | 23.20 | **0.54** | **32.58** | 17.63 | 8.90 | 34.67 |
| D-Flow | 19.67 | 0.29 | 16.44 | 0.27 | 33.65 | 17.19 | 3.69 | 52.28 |
| **FMPlug** | **34.65** | **0.91** | **23.60** | 0.52 | 33.18 | **22.91** | **1.51** | **1.52** |
| Red-diff | 36.55 | 0.98 | 28.71 | 0.62 | 31.59 | 23.77 | 2.05 | 1.85 |

noise. We use 100 samples for evaluation and 5 for guidance; detailed forward models are provided in App. A.6 and Zheng et al. (2025). In terms of methods, we focus on FMPlug, D-Flow, and DIP due to their strong performance in Sec. 4.1. For D-Flow, we choose the best result between random initialization and warm-start using the few-shot instance with the least loss for a fair comparison. We take Red-Diff, one of the SOTA methods with **domain-specific priors** (Zheng et al., 2025), as a reference—performance quoted from their paper.

**Results** From Tab. 4, it is evident that our proposed few-shot FMPlug shows comparable or better performance than DIP and beats D-Flow by a large margin in all the tasks. Qualitatively, from Fig. 6, our method faithfully recovers the main object structures, while D-Flow shows severe artifacts. Moreover, encouragingly, our FMPlug's performance is approaching that of Red-Diff, which relies on domain-specific FM priors for both LIS and BKH. For MRI, we suspect that the considerable gap is due to a slight violation of our assumption on the few-shot guidance: as shown in Fig. 8, these instances are spatially located around a single anatomical position, whereas the evaluation set has a large spatial variability, so many of the evaluation samples possibly do not have any close guidance—violating our assumption. Moreover, we compute the correlation between the learned weights and the ground-truth instance-to-target $\ell_2$ distance across MRI data. We observe a strong negative correlation, with Spearman and Pearson coefficients of $-0.71$ and $-0.79$, respectively. This confirms our intuition: the optimization identifies and assigns larger weights to instances that are structurally closer to the unknown target. Overall, this suggests that with reasonable few-shot guidance, our FMPlug can be a strong candidate method for solving data-sparse scientific IPs.

### 4.3. Ablation study

**Both ingredients of FMPlug are necessary** Tab. 5 shows the performance of FMPlug and two variants: FMPlug-Plain (without warm-start and regularization) and FMPlug-W (with warm-start only). Although both ingredients are necessary for the final performance, most of the perfor-

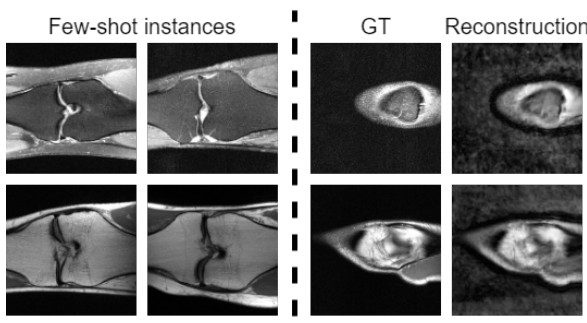

*Figure 8.* Illustration of the concentrated few-shot instances and the under-representative test data.

*Table 5.* Ablation study on **Gaussian Deblur** on DIV2K with additive Gaussian noise ($\sigma = 0.03$). (**Bold**: best, under: second best). **-W**: with warm-start only

|  | PSNR↑ | SSIM↑ | LPIPS↓ | DIST↓ |
|---|---|---|---|---|
| FMPlug-Plain | 25.1602 | 0.6732 | 0.4846 | 0.1719 |
| FMPlug-W | 26.0547 | 0.7193 | 0.4315 | 0.1620 |
| FMPlug | **26.2563** | **0.7339** | **0.4120** | **0.1565** |

mance gain comes from the proposed warm-up strategy. The sharp Gaussianity regularization refines the results.

**FMPlug learns sensible warm-start times** We investigate the behavior of our learnable warm-start time $t^*$ under varying levels of degradation in Gaussian deblurring. Intuitively, the higher the blurring level $\sigma$, the further the observed $y$ from $x$, and hence the smaller $t^*$. Tab. 6 confirms the expected behavior: more severe degradation (larger $\sigma$) yields a smaller $t^*$, pushing the starting point closer to the standard Gaussian prior (the source).

*Table 6.* Learned time $t^*$.

| $\sigma$ | PSNR↑ | $t^*$ |
|---|---|---|
| 1 | 32.24 | 0.83 |
| 3 | 26.58 | 0.78 |
| 6 | 24.13 | 0.72 |

*Table 7.* Sensitivity analysis ($\varepsilon$).

| $\varepsilon$ | PSNR↑ | SSIM↑ | LPIPS↓ |
|---|---|---|---|
| 0 | 26.45 | 0.74 | 0.41 |
| 0.025 | 26.58 | 0.74 | 0.41 |
| 0.1 | 26.44 | 0.74 | 0.41 |

**Sensitivity of the shell thickness and NFE** We perform a sensitivity analysis on the shell thickness $\varepsilon$ in our sharp Gaussian regularization and the number of Function Evaluations (NFE) using the Gaussian deblurring task in the DIV2K dataset. As shown in Tab. 7, $\varepsilon = 0.025$ yields optimal performance. This observation is consistent with our theoretical analysis in Fig. 4: a 2.5% slackness is sufficient to cover the CoM region, balancing constraint sharpness with generative flexibility. Furthermore, Tab. 8 shows that our method is highly robust to low NFEs. Because our warm-start strategy tends to shorten the generation path, it does not require many NFEs to achieve high-fidelity results. Having more NFEs will not only result in diminishing returns and performance saturation but also in significant

memory and computation overheads.

*Table 8.* Sensitivity analysis on the Number of Function Evaluations (NFE).

| NFE | PSNR ↑ | SSIM ↑ | LPIPS ↓ |
|---|---|---|---|
| 1 | 26.44 | 0.74 | 0.41 |
| 3 | 26.58 | 0.74 | 0.41 |
| 5 | 26.62 | 0.74 | 0.41 |

**Runtime and memory** In Tab. 9, we evaluate wall-clock time and peak memory usage for Gaussian deblurring on DIV2K. In general, FlowDPS and FlowChef demonstrate superior computational efficiency. While DIP, D-Flow, and FMPlug are slower, they perform significantly better in terms of recovery quality. D-Flow defaults to PyTorch's built-in `LBFGS` solver for faster convergence; we do not use it here, as it is incompatible with multiple parameter groups. For memory consumption, DIP is the most efficient. D-Flow and FMPlug mitigate the memory bottleneck by applying gradient checkpointing. Overall, interleaving methods (FlowDPS and FlowChef) and plug-in methods (DIP, D-Flow, and FMPlug) show different tradeoffs between recovery quality and speed.

*Table 9.* Wall time and peak GPU memory.

|  | PSNR↑ | Time (s)↓ | Mem (GB)↓ |
|---|---|---|---|
| DIP | 25.23 | 134 | **3.41** |
| FlowChef-P | 20.41 | **7** | 19.41 |
| FlowDPS-P | 20.23 | 10 | 20.89 |
| D-Flow | 23.64 | 456 | 8.91 |
| FMPlug | **26.26** | 492 | 6.84 |

## 5. Conclusion

In this work, we introduced FMPlug, a novel plug-in framework for solving inverse problems (IPs) using foundation flow-matching (FM) models. We identified, *for the first time*, an intriguing and fundamental performance gap between foundation FM priors and domain-specific, and even untrained, priors for solving IPs: **foundation FM models are powerful in generation but weak as generative priors because of their generality**. We took *the first* step to address the gap: our FMPlug framework seamlessly integrates instance guidance and sharp Gaussian regularization with foundation FM priors to maximize priors for IPs. Moreover, we highlighted the relevance of our FMPlug to scientific IPs, which typically concern narrow domains but lack domain-specific FM priors. We also extended our FMPlug framework to the few-shot setting tailored to scientific IP applications and obtained strong performance that often approaches that of domain-specific FM priors. Our work provides a clear, practical path toward transforming foundation FM models into general, reusable generative priors for solving IPs.

## Acknowledgments

This work was partially supported by NSF ACED 2435911 and partially by a UMN DSI Faculty Fellowship. We thank Dr. Ismail Alkhouri (LANL), the anonymous reviewers, and the area chair for their insightful comments on this paper. We acknowledge the Minnesota Supercomputing Institute (MSI) at the University of Minnesota for providing resources that contributed to the research results reported within this paper.

## Impact Statement

The goal of this paper is to leverage pretrained foundation flow matching models to solve inverse problems. This work can be beneficial for solving general image restoration tasks and scientific inverse problems, none of which need to be specifically highlighted here.

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

# A. Appendix

## A.1. Complete algorithms

---

**Algorithm 1** FMPlug

---

**Require:** Measurement $\boldsymbol{y}$, forward model $\mathcal{A}$, FM model $\mathcal{G}_{\boldsymbol{\theta}}$, loss $\ell$, FM schedule functions $\alpha_t, \beta_t$, shell thickness $\varepsilon$, reparameterization $t = \mathrm{sigmoid}(s)$, learning rates $\eta_z, \eta_s$
1: Initialize $\boldsymbol{z}^{(0)} \sim \mathcal{N}(\boldsymbol{0}, \boldsymbol{I})$ and $s^{(0)} = 0$
2: **for** $k = 0, \ldots, N-1$ **do**
3:     $\boldsymbol{x}_{in} \leftarrow \alpha_{\mathrm{sigmoid}(s^{(k)})} \boldsymbol{y} + \beta_{\mathrm{sigmoid}(s^{(k)})} \boldsymbol{z}^{(k)}$     ▷ Construct flow input at current time $t^{(k)} = \mathrm{sigmoid}(s^{(k)})$
4:     $\widehat{\boldsymbol{x}} \leftarrow \mathcal{G}_{\boldsymbol{\theta}}(\boldsymbol{x}_{in}, \mathrm{sigmoid}(s^{(k)}))$     ▷ Simulate flow from $t^{(k)} = \mathrm{sigmoid}(s^{(k)})$ to $1$
5:     $(\boldsymbol{g}_z, g_s) \leftarrow \nabla_{\boldsymbol{z},s} \ell(\boldsymbol{y}, \mathcal{A}(\widehat{\boldsymbol{x}}))$     ▷ Calculate gradients
6:     $s^{(k+1)} \leftarrow s^{(k)} - \eta_s \cdot g_s$     ▷ Update $s$
7:     $\boldsymbol{z}^{(k+1)} \leftarrow \mathcal{P}_{\mathbb{S}_\epsilon^{d-1}(\boldsymbol{0}, \sqrt{d})}(\boldsymbol{z}^{(k)} - \eta_z \cdot \boldsymbol{g}_z)$     ▷ Update $\boldsymbol{z}$
8: **end for**
9: **Return** $\mathcal{G}_{\boldsymbol{\theta}}(\alpha_{t^{(N)}} \boldsymbol{y} + \beta_{t^{(N)}} \boldsymbol{z}^{(N)}, t^{(N)})$

---

---

**Algorithm 2** Few-Shot FMPlug

---

**Require:** Measurement $\boldsymbol{y}$, few-shot set $\{\boldsymbol{x}_j\}_{j=1}^K$, forward model $\mathcal{A}$, FM model $\mathcal{G}_{\boldsymbol{\theta}}$, loss $\ell$, FM schedule functions $\alpha_t, \beta_t$, shell thickness $\varepsilon$, reparameterization $t = \mathrm{sigmoid}(s)$ & $\boldsymbol{w} = \mathrm{softmax}(\boldsymbol{v})$, learning rates $\eta_z, \eta_s, \eta_v$
1: Initialize $\boldsymbol{z}^{(0)} \sim \mathcal{N}(\boldsymbol{0}, \boldsymbol{I})$, $s^{(0)} = 0$, and $\boldsymbol{v}^{(0)} = \boldsymbol{0}$
2: **for** $k = 0, \ldots, N-1$ **do**
3:     $\boldsymbol{w}^{(k)} \leftarrow \mathrm{softmax}(\boldsymbol{v}^{(k)}), \quad \boldsymbol{\mu}^{(k)} \leftarrow \sum_{j=1}^K w_j^{(k)} \boldsymbol{x}_j$     ▷ Compute weights and combination of $\{\boldsymbol{x}_j\}_{j=1}^K$
4:     $\boldsymbol{x}_{in} \leftarrow \alpha_{\mathrm{sigmoid}(s^{(k)})} \boldsymbol{\mu}^{(k)} + \beta_{\mathrm{sigmoid}(s^{(k)})} \boldsymbol{z}^{(k)}$     ▷ Construct flow input at current time $t^{(k)} = \mathrm{sigmoid}(s^{(k)})$
5:     $\widehat{\boldsymbol{x}} \leftarrow \mathcal{G}_{\boldsymbol{\theta}}(\boldsymbol{x}_{in}, \mathrm{sigmoid}(s^{(k)}))$     ▷ Simulate flow from $t^{(k)} = \mathrm{sigmoid}(s^{(k)})$ to $1$
6:     $(\boldsymbol{g}_z, g_s, \boldsymbol{g}_v) \leftarrow \nabla_{\boldsymbol{z},s,\boldsymbol{v}} \ell(\boldsymbol{y}, \mathcal{A}(\widehat{\boldsymbol{x}}))$     ▷ Calculate gradients
7:     $s^{(k+1)} \leftarrow s^{(k)} - \eta_s \cdot g_s; \quad \boldsymbol{v}^{(k+1)} \leftarrow \boldsymbol{v}^{(k)} - \eta_v \cdot \boldsymbol{g}_v$     ▷ Update $s$ and $\boldsymbol{v}$
8:     $\boldsymbol{z}^{(k+1)} \leftarrow \mathcal{P}_{\mathbb{S}_\epsilon^{d-1}(\boldsymbol{0}, \sqrt{d})}(\boldsymbol{z}^{(k)} - \eta_z \cdot \boldsymbol{g}_z)$     ▷ Update $\boldsymbol{z}$
9: **end for**
10: $\boldsymbol{w}^{(N)} \leftarrow \mathrm{softmax}(\boldsymbol{v}^{(N)}), \quad \boldsymbol{\mu}^{(N)} \leftarrow \sum_{j=1}^K w_j^{(N)} \boldsymbol{x}_j$     ▷ Compute final combination
11: **Return** $\mathcal{G}_{\boldsymbol{\theta}}(\alpha_{t^{(N)}} \boldsymbol{\mu}^{(N)} + \beta_{t^{(N)}} \boldsymbol{z}^{(N)}, t^{(N)})$

---

## A.2. Details about the image regression experiment in Tab. 2

In this task, we solve $\min_{\boldsymbol{z}} \ell(\boldsymbol{x}, \mathcal{G}_{\boldsymbol{\theta}}(\boldsymbol{z}))$, i.e., regressing any given image $\boldsymbol{x}$ with $\mathcal{G}_{\boldsymbol{\theta}}$. The purpose is to test whether $\mathcal{G}_{\boldsymbol{\theta}}$ is powerful enough to generate arbitrary given natural images, i.e., its surjectivity. We use 1000 randomly drawn images from the training set of `DIV2K` and adopt all default hyperparameters from App. A.3. For D-Flow, we stop the optimization process when there is no effective update to $\boldsymbol{z}$ for 5 consecutive epochs. We run FMPlug for a maximum of 1000 epochs and take the generation results from the last epoch.

## A.3. Experiment details for Sec. 4.1

In this section, we provide implementation details on all methods compared in Sec. 4.1. By default, we use Stable Diffusion V3 Medium[3] (Patrick Esser et al, 2024) as the backbone model whenever foundation FM models are needed.

- **FMPlug** We use `AdamW` as our default optimizer; The number of function evaluations (NFE) is 3 and we use the `Heun2` ODE solver to balance efficiency and accuracy; The learning rate for $\boldsymbol{z}$ is 0.5, and for $s$ is 0.005.
- **D-Flow** We use their default optimizer: `LBFGS` with line search; NFE $= 6$ with the `Heun2` ODE solver; regularization strength $\lambda = 0.01$; We perform the initialization with the Euler ODE solver.
- **FlowDPS** We set NFE $= 28$ with `FlowMatchEulerDiscreteScheduler`; For their data consistency term, we perform

---

[3] https://huggingface.co/stabilityai/stable-diffusion-3-medium

3 steps of gradient descent (i.e., lazy optimization) with `step size` = 15.0.

- **FlowChef** We set NFE = 28 with `FlowMatchEulerDiscreteScheduler`; We use `step size` = 50.0 for simple-distortion tasks.
- **Deep Image Prior** We use a 5-layer UNet with 256 channels for each layer with `AdamW` optimizer; We set the learning rate for the network to 0.001.

For D-Flow and FMPlug, to save memory, we apply the gradient checkpointing technique.

## A.4. Complete results for Sec. 4.1

*Table 10.* **Inpainting** and $4\times$ Super Resolution on AFHQ with additive Gaussian noise ($\sigma = 0.03$). (**Bold**: best, under: second best)

| | Inpainting | | | | | | Super Resolution $4\times$ | | | | | |
|---|---|---|---|---|---|---|---|---|---|---|---|---|
| method | PSNR ↑ | SSIM ↑ | LPIPS ↓ | DISTS ↓ | CLIPIQA ↑ | MUSIQ ↑ | PSNR ↑ | SSIM ↑ | LPIPS ↓ | DISTS ↓ | CLIPIQA ↑ | MUSIQ ↑ |
| DIP | **33.32** | **0.90** | **0.21** | 0.07 | 0.47 | 57.73 | 29.85 | 0.78 | 0.37 | **0.12** | 0.33 | 43.38 |
| FlowChef-P | 29.27 | 0.77 | 0.41 | 0.21 | 0.57 | 36.48 | 29.23 | 0.79 | 0.38 | 0.19 | 0.64 | 38.77 |
| FlowChef | 29.35 | 0.77 | 0.41 | 0.21 | 0.58 | 37.02 | 29.25 | 0.79 | 0.38 | 0.19 | **0.65** | 39.01 |
| FlowDPS-P | 27.63 | 0.73 | 0.41 | 0.17 | 0.43 | 56.70 | 28.75 | 0.76 | 0.37 | 0.15 | 0.37 | 52.74 |
| FlowDPS | 27.53 | 0.72 | 0.47 | 0.18 | 0.35 | 49.14 | 28.60 | 0.75 | 0.42 | 0.16 | 0.35 | 47.61 |
| D-Flow | 28.43 | 0.76 | 0.41 | 0.17 | 0.65 | 60.45 | 26.37 | 0.70 | 0.54 | 0.18 | 0.31 | **53.13** |
| **FMPlug-W** | 32.75 | 0.88 | 0.37 | 0.08 | 0.63 | 60.87 | 30.13 | **0.81** | 0.34 | 0.13 | 0.18 | 47.43 |
| **FMPlug** | 32.81 | 0.87 | 0.34 | **0.06** | **0.66** | **61.86** | **30.31** | 0.81 | **0.33** | **0.12** | 0.20 | 46.91 |

*Table 11.* **Gaussian Blur** and **Motion Blur** on AFHQ with additive Gaussian noise ($\sigma = 0.03$). (**Bold**: best, under: second best)

| | Gaussian Blur | | | | | | Motion Blur | | | | | |
|---|---|---|---|---|---|---|---|---|---|---|---|---|
| method | PSNR ↑ | SSIM ↑ | LPIPS ↓ | DISTS ↓ | CLIPIQA ↑ | MUSIQ ↑ | PSNR ↑ | SSIM ↑ | LPIPS ↓ | DISTS ↓ | CLIPIQA ↑ | MUSIQ ↑ |
| DIP | 29.39 | 0.77 | **0.39** | 0.14 | 0.30 | 36.07 | 28.69 | 0.75 | 0.38 | 0.16 | 0.26 | 34.88 |
| FlowChef-P | 23.84 | 0.63 | 0.54 | 0.30 | 0.28 | 15.81 | 24.77 | 0.66 | 0.50 | 0.28 | **0.37** | 19.99 |
| FlowChef | 23.87 | 0.63 | 0.54 | 0.30 | 0.28 | 15.89 | 24.78 | 0.66 | 0.50 | 0.28 | **0.37** | 19.86 |
| FlowDPS-P | 24.15 | 0.60 | 0.49 | 0.24 | 0.23 | 42.74 | 24.81 | 0.64 | 0.46 | 0.21 | 0.28 | 47.77 |
| FlowDPS | 23.69 | 0.58 | 0.55 | 0.27 | 0.15 | 30.28 | 24.49 | 0.62 | 0.52 | 0.24 | 0.20 | 36.63 |
| D-Flow | 25.90 | 0.66 | 0.54 | 0.20 | **0.34** | **50.61** | 27.81 | 0.73 | 0.48 | 0.17 | 0.35 | 47.74 |
| **FMPlug-W** | 30.38 | **0.79** | 0.40 | **0.12** | 0.22 | 42.02 | 30.10 | 0.79 | 0.39 | 0.12 | 0.26 | 48.62 |
| **FMPlug** | **30.41** | 0.79 | **0.39** | 0.12 | 0.21 | 43.08 | **30.43** | **0.81** | **0.37** | **0.11** | 0.28 | **52.23** |

*Table 12.* **Inpainting** and $4\times$ Super Resolution on DIV2K with additive Gaussian noise ($\sigma = 0.03$). (**Bold**: best, under: second best)

| | Inpainting | | | | | | Super Resolution $4\times$ | | | | | |
|---|---|---|---|---|---|---|---|---|---|---|---|---|
| method | PSNR ↑ | SSIM ↑ | LPIPS ↓ | DISTS ↓ | CLIPIQA ↑ | MUSIQ ↑ | PSNR ↑ | SSIM ↑ | LPIPS ↓ | DISTS ↓ | CLIPIQA ↑ | MUSIQ ↑ |
| DIP | 28.49 | **0.86** | **0.27** | 0.09 | 0.59 | 55.82 | 25.75 | 0.73 | 0.42 | **0.15** | 0.40 | 37.85 |
| FlowChef-P | 24.67 | 0.67 | 0.46 | 0.24 | 0.50 | 38.04 | 25.08 | 0.71 | 0.43 | 0.22 | **0.60** | 38.50 |
| FlowChef | 24.76 | 0.67 | 0.46 | 0.24 | 0.50 | 38.87 | 25.09 | 0.71 | 0.43 | 0.22 | **0.60** | 38.67 |
| FlowDPS-P | 24.01 | 0.65 | 0.47 | 0.19 | 0.54 | 49.49 | 24.92 | 0.69 | 0.42 | 0.17 | 0.51 | 47.19 |
| FlowDPS | 24.04 | 0.64 | 0.50 | 0.19 | 0.47 | 46.89 | 24.83 | 0.68 | 0.45 | 0.17 | 0.46 | 44.80 |
| D-Flow | 24.71 | 0.73 | 0.41 | 0.18 | 0.67 | 62.25 | 23.42 | 0.64 | 0.52 | 0.17 | 0.37 | **57.18** |
| **FMPlug-W** | 28.82 | 0.85 | 0.33 | 0.08 | 0.68 | **65.09** | 25.77 | **0.74** | **0.38** | **0.15** | 0.24 | 40.96 |
| **FMPlug** | **28.95** | 0.84 | 0.32 | **0.07** | **0.69** | 64.80 | **25.88** | **0.74** | **0.38** | **0.15** | 0.27 | 40.30 |

*Table 13.* **Gaussian Blur** and **Motion Blur** on DIV2K with additive Gaussian noise ($\sigma = 0.03$). (**Bold**: best, under: second best)

| method | Gaussian Blur | | | | | | Motion Blur | | | | | |
| --- | --- | --- | --- | --- | --- | --- | --- | --- | --- | --- | --- | --- |
| | PSNR ↑ | SSIM ↑ | LPIPS ↓ | DISTS ↓ | CLIPIQA ↑ | MUSIQ ↑ | PSNR ↑ | SSIM ↑ | LPIPS ↓ | DISTS ↓ | CLIPIQA ↑ | MUSIQ ↑ |
| DIP | 25.23 | 0.70 | 0.43 | 0.18 | **0.38** | 32.54 | 24.75 | 0.68 | 0.45 | 0.20 | 0.35 | 32.59 |
| FlowChef-P | 20.41 | 0.49 | 0.62 | 0.34 | 0.23 | 16.68 | 21.27 | 0.54 | 0.57 | 0.32 | 0.34 | 19.76 |
| FlowChef | 20.41 | 0.49 | 0.62 | 0.34 | 0.23 | 16.68 | 21.28 | 0.54 | 0.57 | 0.32 | 0.34 | 19.82 |
| FlowDPS-P | 20.23 | 0.46 | 0.58 | 0.29 | 0.32 | 35.90 | 21.07 | 0.51 | 0.54 | 0.26 | 0.39 | 39.56 |
| FlowDPS | 20.22 | 0.45 | 0.61 | 0.30 | 0.20 | 30.51 | 21.05 | 0.50 | 0.58 | 0.27 | 0.26 | 34.21 |
| D-Flow | 23.64 | 0.64 | 0.52 | 0.17 | 0.37 | **53.03** | 25.21 | 0.70 | 0.47 | 0.17 | **0.42** | **53.78** |
| **FMPlug-W** | 26.05 | 0.72 | 0.43 | **0.16** | 0.29 | 36.66 | 26.83 | 0.74 | 0.40 | 0.14 | 0.36 | 46.95 |
| **FMPlug** | **26.26** | **0.73** | **0.41** | **0.16** | 0.28 | 38.14 | **27.38** | **0.78** | **0.36** | **0.12** | **0.42** | 51.71 |

*Table 14.* **Inpainting** and 4× Super Resolution on RealSR with additive Gaussian noise ($\sigma = 0.03$). (**Bold**: best, under: second best)

| method | Inpainting | | | | | | Super Resolution 4× | | | | | |
| --- | --- | --- | --- | --- | --- | --- | --- | --- | --- | --- | --- | --- |
| | PSNR ↑ | SSIM ↑ | LPIPS ↓ | DISTS ↓ | CLIPIQA ↑ | MUSIQ ↑ | PSNR ↑ | SSIM ↑ | LPIPS ↓ | DISTS ↓ | CLIPIQA ↑ | MUSIQ ↑ |
| DIP | 30.88 | **0.89** | **0.25** | 0.09 | 0.47 | 54.97 | **26.81** | 0.72 | 0.44 | **0.17** | 0.30 | 38.23 |
| FlowChef-P | 25.81 | 0.69 | 0.45 | 0.25 | 0.35 | 35.96 | 25.89 | 0.71 | 0.43 | 0.24 | **0.44** | 35.42 |
| FlowChef | 25.89 | 0.69 | 0.45 | 0.25 | 0.35 | 36.61 | 25.92 | 0.71 | 0.43 | 0.23 | **0.44** | 35.65 |
| FlowDPS-P | 25.68 | 0.69 | 0.47 | 0.20 | 0.36 | 49.28 | 26.11 | 0.71 | 0.43 | 0.18 | 0.34 | 46.24 |
| FlowDPS | 25.78 | 0.69 | 0.48 | 0.19 | 0.32 | 46.54 | 26.10 | 0.70 | 0.45 | 0.18 | 0.32 | 44.49 |
| D-Flow | 25.27 | 0.69 | 0.42 | 0.21 | **0.59** | 60.99 | 23.60 | 0.62 | 0.53 | 0.20 | 0.28 | **56.53** |
| **FMPlug-W** | 31.30 | 0.88 | 0.28 | 0.07 | 0.56 | **62.77** | 26.58 | 0.73 | 0.39 | **0.17** | 0.16 | 40.05 |
| **FMPlug** | **31.79** | **0.89** | 0.26 | **0.06** | 0.56 | 62.61 | 26.66 | **0.74** | **0.38** | **0.17** | 0.17 | 39.27 |

*Table 15.* **Gaussian Blur** and **Motion Blur** on RealSR with additive Gaussian noise ($\sigma = 0.03$). (**Bold**: best, under: second best)

| method | Gaussian Blur | | | | | | Motion Blur | | | | | |
| --- | --- | --- | --- | --- | --- | --- | --- | --- | --- | --- | --- | --- |
| | PSNR ↑ | SSIM ↑ | LPIPS ↓ | DISTS ↓ | CLIPIQA ↑ | MUSIQ ↑ | PSNR ↑ | SSIM ↑ | LPIPS ↓ | DISTS ↓ | CLIPIQA ↑ | MUSIQ ↑ |
| DIP | 26.17 | 0.70 | 0.46 | 0.20 | 0.28 | 31.78 | 26.17 | 0.70 | 0.46 | 0.22 | 0.28 | 33.25 |
| FlowChef-P | 21.42 | 0.51 | 0.63 | 0.36 | 0.19 | 16.65 | 22.50 | 0.56 | 0.56 | 0.33 | 0.26 | 20.77 |
| FlowChef | 21.42 | 0.51 | 0.63 | 0.36 | 0.19 | 16.68 | 22.51 | 0.56 | 0.56 | 0.33 | 0.26 | 20.88 |
| FlowDPS-P | 21.21 | 0.47 | 0.59 | 0.30 | 0.22 | 38.23 | 22.50 | 0.55 | 0.55 | 0.27 | 0.27 | 41.01 |
| FlowDPS | 21.21 | 0.47 | 0.61 | 0.31 | 0.17 | 33.68 | 22.55 | 0.54 | 0.56 | 0.28 | 0.22 | 37.84 |
| D-Flow | 23.65 | 0.60 | 0.54 | 0.20 | **0.30** | **54.62** | 25.86 | 0.69 | 0.47 | 0.21 | **0.31** | **51.57** |
| **FMPlug-W** | 27.05 | 0.72 | 0.44 | **0.18** | 0.21 | 34.47 | 28.01 | 0.76 | 0.40 | 0.16 | 0.28 | 43.86 |
| **FMPlug** | **27.22** | **0.73** | **0.43** | **0.18** | 0.19 | 36.00 | **28.63** | **0.79** | **0.37** | **0.14** | 0.30 | 48.07 |

As noted in Sec. 4.1, the proposed method does not consistently achieve the highest scores on CLIPIQA and MUSIQ. Although these no-reference metrics are widely used in the super-resolution community to assess perceptual quality in the absence of ground-truth images (Sun et al., 2025; Wu et al., 2024a;b), they must be interpreted with caution. Recent studies (Rasheed et al., 2026; Liao et al., 2026; Antsiferova et al., 2024) have raised significant concerns about their robustness, particularly highlighting their vulnerability to adversarial attacks. Consequently, we argue that these metrics should primarily be considered when full-reference evaluations yield comparable results. Therefore, in this work, we strictly treat no-reference metrics as complementary indicators.

### A.5. Domain-specific prior vs foundation prior

To benchmark our progress in bridging the performance gap between foundation and domain-specific priors, we expand Tab. 1 to include more competing methods with domain-specific (DS) FM priors in Tab. 16. In particular, we include the recent OT-ODE (Pokle et al., 2023), PnP-Flow (Martin et al., 2025) based on a pretrained FM model on AFHQ-Cat from (Martin et al., 2025). We note in passing that OT-ODE and PnP-Flow are not compatible with latent FM models, and

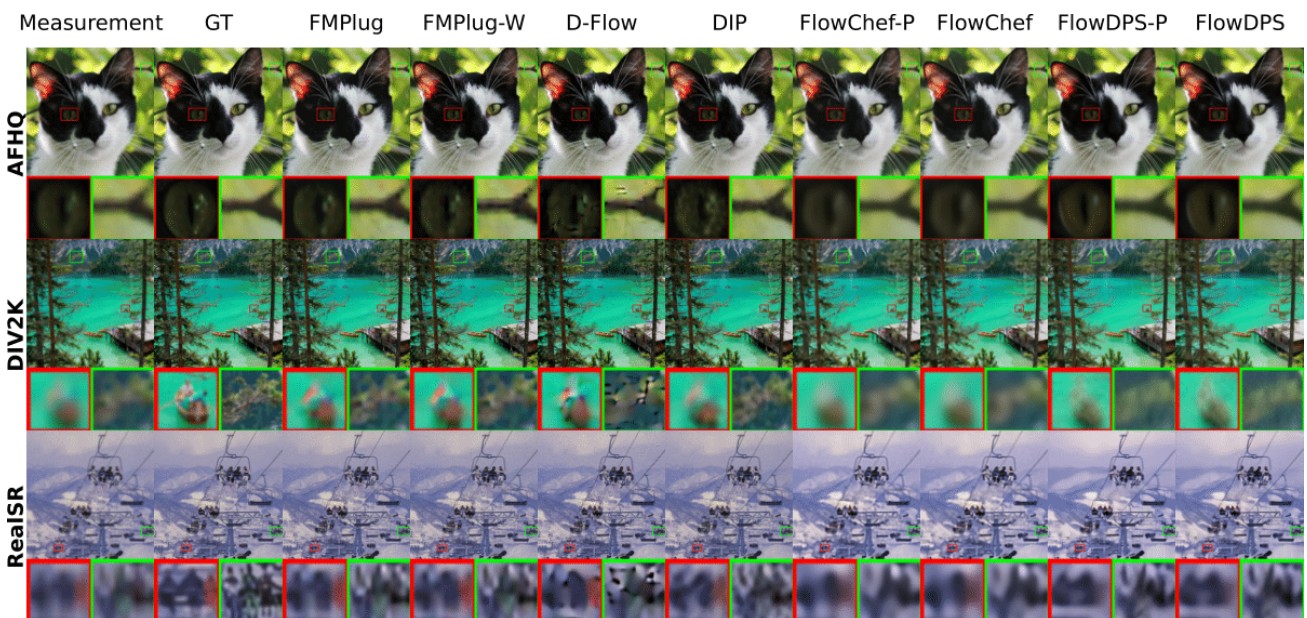

*Figure 9.* Qualitative comparison in $4\times$ super resolution task.

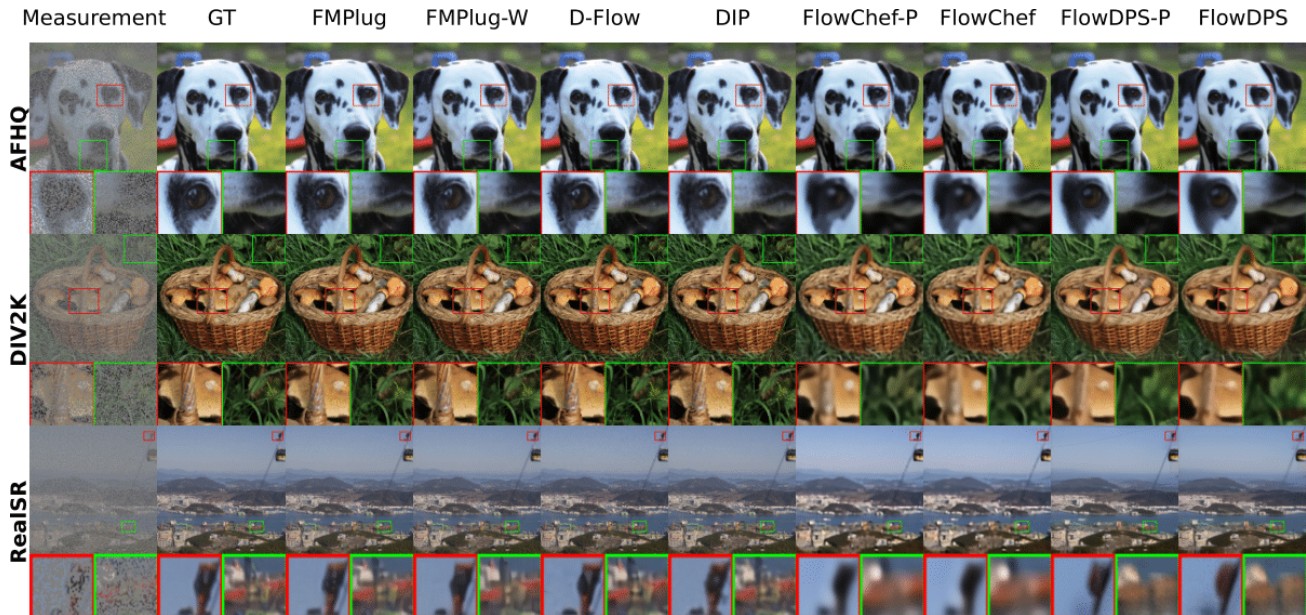

*Figure 10.* Qualitative comparison in Inpainting task.

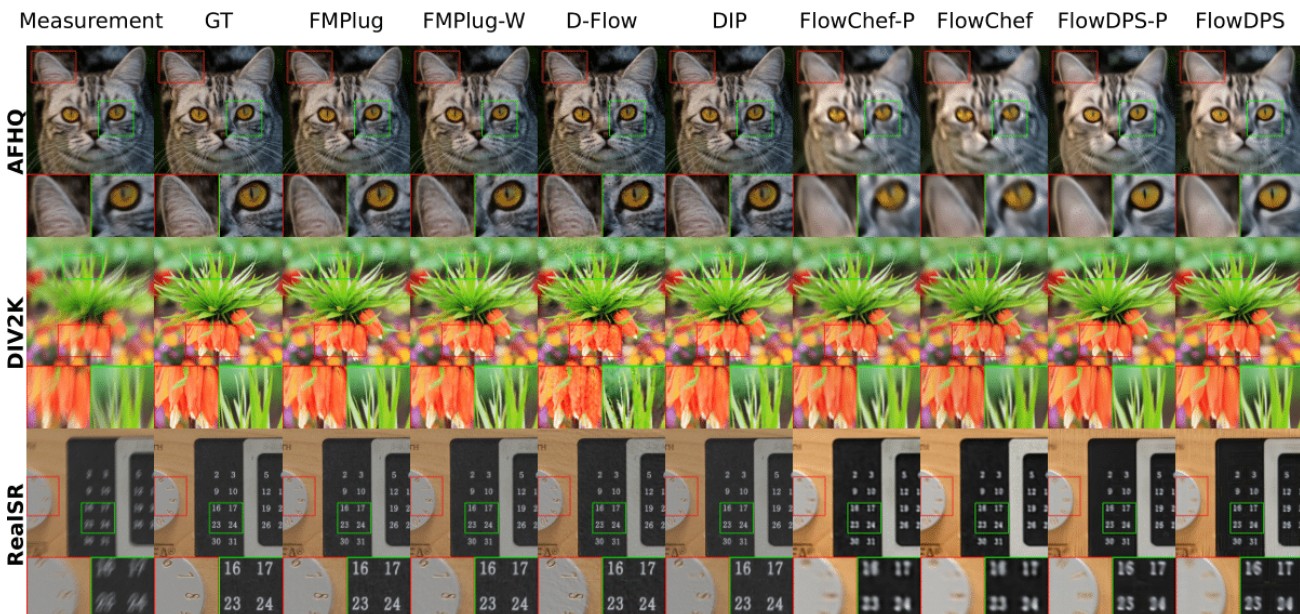

*Figure 11.* Qualitative comparison in motion deblur task.

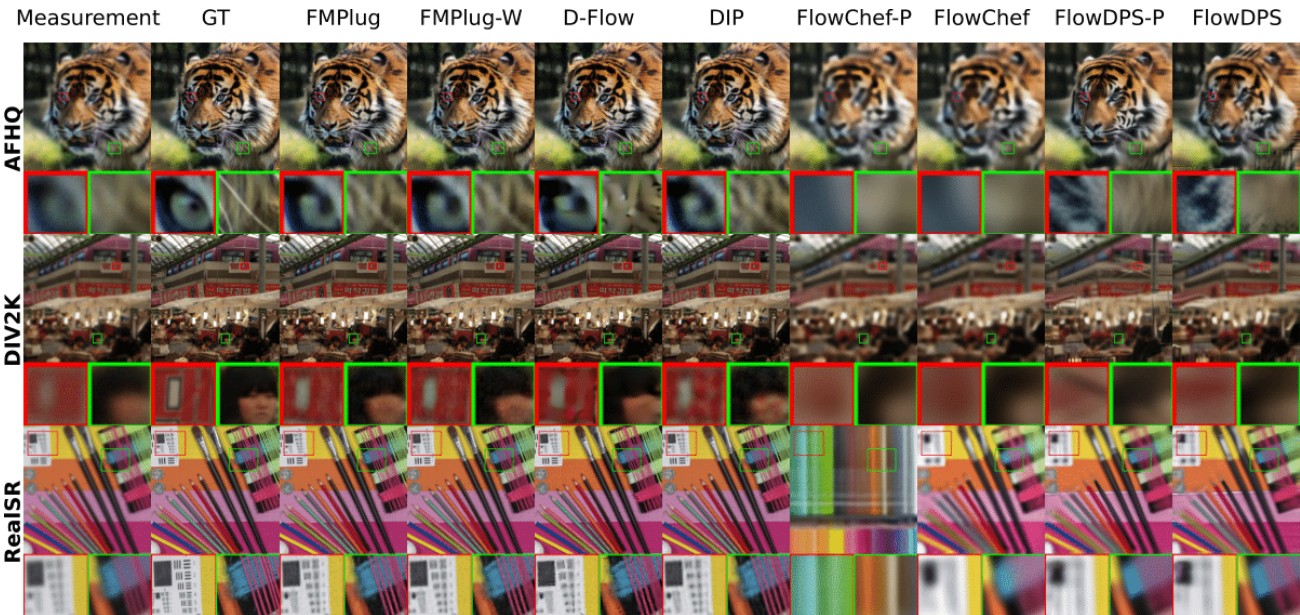

*Figure 12.* Qualitative comparison in Gaussian deblur task.

*Table 16.* **Gaussian Deblur** and $4\times$ Super Resolution on AFHQ-Cat $256 \times 256$ with additive Gaussian noise ($\sigma = 0.03$). FD: Foundation; DS: Domain-specific; **Bold**: best, under: second best; -: not available

| | Super Resolution $4\times$ | | | | | | Gaussian Blur | | | | | |
|---|---|---|---|---|---|---|---|---|---|---|---|---|
| | LPIPS↓ | PSNR↑ | SSIM↑ | DIST↓ | CLIPIQA↑ | MUSIQ↑ | LPIPS↓ | PSNR↑ | SSIM↑ | DIST↓ | CLIPIQA↑ | MUSIQ↑ |
| DIP | 0.36 | 28.17 | 0.76 | 0.21 | 0.25 | 28.12 | 0.36 | 27.92 | 0.75 | 0.23 | 0.26 | 23.94 |
| OT-ODE (DS) | **0.19** | 26.43 | 0.74 | 0.90 | 0.59 | **64.63** | **0.19** | 27.67 | 0.75 | 0.89 | 0.62 | **63.82** |
| OT-ODE (FD) | - | - | - | - | - | - | - | - | - | - | - | - |
| PnP-Flow (DS) | 0.24 | 27.45 | **0.80** | 0.82 | 0.52 | 51.95 | 0.31 | 28.70 | **0.79** | 0.77 | **0.66** | 40.26 |
| PnP-Flow (FD | - | - | - | - | - | - | - | - | - | - | - | - |
| FlowDPS (DS) | 0.24 | 28.56 | 0.79 | 0.14 | 0.57 | 55.63 | 0.38 | 22.27 | 0.56 | **0.20** | 0.52 | 52.42 |
| FlowDPS (FD) | 0.37 | 24.45 | 0.74 | 0.27 | **0.63** | 27.96 | 0.55 | 22.11 | 0.59 | 0.38 | 0.28 | 15.10 |
| D-Flow (DS) | 0.27 | 25.81 | 0.69 | 0.82 | 0.52 | 57.74 | 0.20 | 28.41 | 0.77 | 0.87 | 0.61 | 59.29 |
| D-Flow (FD) | 0.53 | 24.64 | 0.67 | 0.31 | 0.31 | 45.27 | 0.56 | 24.42 | 0.62 | 0.21 | 0.30 | 49.12 |
| **FMPlug (DS)** | 0.22 | 27.52 | 0.79 | **0.12** | 0.61 | 61.21 | 0.36 | 27.44 | 0.75 | 0.77 | 0.24 | 31.19 |
| **FMPlug (FD)** | 0.33 | **28.85** | **0.80** | 0.22 | 0.31 | 28.77 | 0.35 | **29.00** | **0.79** | 0.23 | 0.24 | 30.58 |

hence we do not report their performance with FD FM priors. On both Gaussian deblurring and super-resolution, by most of the metrics, our FMPlug (FD) gets closer or even comparable to these methods with DS FM priors in performance.

### A.6. Details of scientific IPs in Sec. 4.2

Our exposition below follows the descriptions in Section 3 and Appendix B of Zheng et al. (2025).

**Linear inverse scattering** Inverse scattering is an IP in optical microscopy, where the objective is to recover an unknown permittivity contrast field $z \in \mathbb{R}^n$ from the scattered light field $y_{\text{sc}} \in \mathbb{C}^m$:

$$y_{\text{sc}} = H(u_{\text{tot}} \odot z) + n \in \mathbb{C}^m \quad \text{where} \quad u_{\text{tot}} = G(u_{\text{in}} \odot z). \tag{A.1}$$

Here, $G \in \mathbb{C}^{n \times n}$ and $H \in \mathbb{C}^{m \times n}$ denote the discretized Green's functions that characterize the response of the optical system, $u_{\text{in}}$ and $u_{\text{tot}}$ are the incident and total lightfields, $\odot$ represents the elementwise (Hadamard) product, and $n$ accounts for measurement noise. This model is nonlinear in $z$ as $y_{\text{sc}} = H(G(u_{\text{in}} \odot z) \odot z) + n$. Under the first Born approximation, the scattering is assumed to be weak enough so that the total field is roughly equal to the input field, i.e., $u_{\text{tot}} \approx u_{\text{in}}$, leading to the linear forward model

$$y_{\text{sc}} \approx H(u_{\text{in}} \odot z) + n \in \mathbb{C}^m. \tag{A.2}$$

The resolution of the LIS data is $(1, 128, 128)$.

**Compressed sensing MRI** Compressed sensing MRI (CS-MRI) is an important technique to accelerate MRI scanning via subsampling. We consider the parallel imaging (PI) variant, which can be formulated as recovering an unknown complex-valued image $x \in \mathbb{C}^n$ from a set of measurements

$$y_j = \mathcal{P}\mathcal{F}S_j x + n_j \quad \text{for } j = 1, \dots, J \tag{A.3}$$

where $\mathcal{P} \in \{0, 1\}^{m \times n}$ and $\mathcal{F}$ are the sub-sampling and Fourier operators, respectively; and $y_j$, $S_j$, and $n_j$ are the measurement, sensitivity map, and noise corresponding to the $j$-th coil, for $j = 1, \dots, J$. The resolution of the target MRI image is $(2, 320, 320)$, where the 2 channels are for the real and imaginary parts, respectively.

**Black hole imaging** Black Hole (BKH) imaging relies on Very Long Baseline Interferometry (VLBI) to acquire data. In this process, each pair of telescopes $(a, b)$ provides a measure $y_{a,b}^t$ of the ideal visibility $v_{a,b}^t$, the latter sampling a specific spatial Fourier frequency of the source image corresponding to the projected baseline at time $t$:

$$y_{a,b}^t = g_a^t g_b^t e^{-i(\phi_a^t - \phi_b^t)} v_{a,b}^t(z) + \eta_{a,b}^t. \tag{A.4}$$

Here, $g_a^t, g_b^t$ are unknown real-valued amplitude gains for telescopes $a$ and $b$, and the unknown phase factor $e^{-i(\phi_a^t - \phi_b^t)}$ represents the relative phase shift due to atmospheric turbulence and calibration errors. $\eta_{a,b}^t$ models complex-valued Gaussian thermal noise.

In practice, these significantly distorted visibility measurements are not directly used for recovery due to the highly nonlinear nature of the forward model. Instead, the synthetic *closure phases* and *(log) closure amplitudes* are calculated that eliminate the influence of the phase shift and amplitude gains, respectively:

$$y_{t,(a,b,c)}^{\mathrm{cp}} = \angle(V_{a,b}^t V_{b,c}^t V_{a,c}^t) \in \mathbb{R}, \quad y_{t,(a,b,c,d)}^{\mathrm{logca}} = \log |V_{a,b}^t||V_{c,d}^t|/|V_{a,c}^t||V_{b,d}^t| \in \mathbb{R}, \tag{A.5}$$

where $\angle$ represents the complex angle and $|\cdot|$ the amplitude. For an array of $M$ telescopes, at time $t$, there are $(M-1)(M-2)/2$ independent closure phase measurements and $M(M-3)/2$ log closure amplitude measurements. In addition, the DC component of the Fourier transform, i.e., the total flux of the image source, is measured separately. The IP of the BKH imaging is to estimate a $z$ that is consistent with all the closure phase, closure amplitude, and DC measurements.

The spatial resolution of the BKH images is $(64, 64)$. These images can be either real-valued, i.e., single-channel, or complex-valued, i.e., two-channel. To address the mismatch of these channel numbers with that of the foundation generative prior, which is always 3—RGB channels, we implement a simple replication strategy: (1) for single-channel data, we replicate the single-channel image into three channels when setting up the optimization loss; for reconstruction, we extract a single channel from the 3-channel generation result; (2) for two-channel data, we map the data to the first two channels and zero-fill the third channel when setting up the optimization loss; for reconstruction, we simply discard the third channel.

### A.7. Additional ablation study

**Robustness to random initializations**  We perform Gaussian deblurring on a subset of the DIV2K dataset, with 10 random initializations per image. As detailed in Tab. 17, our method shows remarkably lower variances than FlowDPS, FlowChef, and D-Flow on all metrics. This high stability eliminates the need for computationally expensive multiple restarts in practice.

*Table 17.* Performance and variance across 10 random initializations for Gaussian deblurring.

| Method | PSNR ↑ | SSIM ↑ | LPIPS ↓ |
|---|---|---|---|
| FlowDPS | 20.65 (±0.163) | 0.47 (±0.110) | 0.58 (±0.017) |
| FlowChef | 20.77 (±0.047) | 0.49 (±0.001) | 0.62 (±0.002) |
| D-Flow | 24.43 (±0.625) | 0.65 (±0.025) | 0.50 (±0.028) |
| Ours | **26.60 (±0.025)** | **0.74 (±0.001)** | **0.40 (±0.004)** |

**Sensitivity of the few-shot setting to the number of instances**  We vary the number of available few-shot instances $(k)$ out of a total of 6 in the MRI reconstruction task. From Tab. 18, we observe a clear improvement in performance as $k$ increases. In particular, leveraging only a single instance $(k = 1)$ already provides a substantial improvement over the zero-shot baseline $(k = 0)$, demonstrating the extreme data efficiency of our method.

*Table 18.* Ablation on the number of few-shot instances $(k)$ for MRI reconstruction.

| $k$ Instances | PSNR ↑ | SSIM ↑ | Data Misfit ↓ |
|---|---|---|---|
| 0 (Zero-shot) | 16.36 | 0.39 | 37.66 |
| 1 | 22.22 | 0.48 | 34.17 |
| 3 | 22.36 | 0.49 | 33.97 |
| Full (6) | 23.60 | 0.52 | 33.18 |

