# OpenReview forum: "Saving Foundation Flow-Matching Priors for Inverse Problems"
_ICML.cc/2026/Conference — ICML 2026 regular_

### Official Review · Reviewer_1Nvs · 2026-02-23

**Soundness:** 3
**Presentation:** 3
**Significance:** 3
**Originality:** 2
**Overall Recommendation:** 5
**Confidence:** 4

**Summary:**

This paper identifies and addresses a practical gap: foundation flow-matching (FM) models (e.g., Stable Diffusion V3), when used as priors for inverse problems (IPs), substantially underperform domain-specific FM priors and even untrained priors like Deep Image Prior (DIP). To close this gap, the authors propose FMPlug, a plug-in framework with two key ingredients: (1) an instance-guided, time-dependent warm-start that injects the measurement $y$ (or a weighted combination of few-shot instances) at a learnable time point $t$ along the FM flow, rather than starting from $t=0$; and (2) a sharp Gaussianity regularization that constrains the latent $z$ to lie on a thin shell $\mathbb{S}^{d-1}_\varepsilon(0, \sqrt{d})$ via closed-form projection, replacing the flat $\chi^2$ log-likelihood regularizer used in D-Flow. Experiments span four image restoration tasks on three datasets and three scientific IPs from InverseBench (LIS, MRI, black hole imaging), demonstrating consistent improvements over D-Flow and interleaving baselines.

**Compliance With Llm Reviewing Policy:**

Affirmed.

**Key Questions For Authors:**

1. What is the distribution of learned $t^* $ values across tasks and difficulty levels? Does $t^* $ increase with degradation severity as the theory in Sec. 3.1 would predict?
2. How sensitive is performance to the shell thickness $\varepsilon$? What happens at $\varepsilon = 0$ (hard sphere) vs. $\varepsilon = 0.1$?
3. For the few-shot scientific IPs, what do the learned weights $w^*$ look like? Do they indeed concentrate on the instance closest to the ground truth?
4. Have you tried multiple random restarts for the $(z, t)$ optimization? How much variance is there across runs for a single test image?
5. For super-resolution, how exactly is $y$ (at $128 \times 128$) used in the warm-start $\alpha_t y + \beta_t z$ when the latent space expects a $512 \times 512$-equivalent input?
6. Can you provide an ablation on the number of few-shot instances $K$ for the scientific IPs? How does performance degrade with $K = 1, 2, 3, ...$?

**Limitations:**

The Impact Statement is minimal but acceptable for a methods paper. However, the paper does not adequately discuss its own limitations:

- The **simple-distortion assumption** ($x \approx y$) is central to the warm-start but its failure modes are not characterized. When does the approximation $z_t \approx \alpha_t y + \beta_t z$ become too coarse?
- The **few-shot assumption** that guidance instances are close to the target is violated in the MRI experiment (Fig. 8), and the authors note this but do not discuss how a practitioner would know *a priori* whether this assumption holds.
- The **computational cost** (~8 min/image) is not framed as a limitation, despite being a significant practical barrier.
- The **non-convex optimization** landscape and potential for local minima are not discussed.
- The method's **systematic underperformance on perceptual metrics** (CLIPIQA, MUSIQ) is rationalized but not analyzed as a limitation.

**Suggestion:** Add a Limitations paragraph discussing: (i) when the closeness assumptions fail, (ii) computational overhead, (iii) the perception-distortion tradeoff, and (iv) sensitivity to few-shot instance quality.

**Strengths And Weaknesses:**

## Strengths

**1. Well-motivated problem identification.**
The paper clearly articulates why foundation FM priors underperform for IPs — they constrain the object only to be a natural image, not a domain-specific one — and provides compelling empirical evidence (Tab. 1, Fig. 3) that this gap is real and consistent across difficulty levels. The observation that existing strengthening techniques (D-Flow's initialization, FlowDPS's text prompts) are essentially ineffective (Tab. 1, FD vs FD-S rows) is a valuable diagnostic contribution.

**2. Clean and well-grounded warm-start strategy.**
The time-dependent warm-start (Eq. 3.5) is well-motivated through the FM flow structure. The argument that D-Flow's initialization $z_0 = \sqrt{\alpha} y_0 + \sqrt{1-\alpha} z$ places the seed in $\mathcal{S}^c$ (the complement of the Gaussian concentration shell) with high probability is a crisp application of concentration-of-measure (Thm. 3.1), and the proposed fix — starting at a learnable $t > 0$ along the flow with $y$ substituting for $x$ — is natural and elegant. Making $t$ learnable rather than fixed is a good design choice, as it automatically trades off approximation error $\alpha_t \varepsilon$ against flow traversal length.

**3. Principled Gaussianity regularization.**
The critique of D-Flow's $\chi^2$ negative log-likelihood regularizer is insightful and well-visualized (Fig. 4): the function is essentially flat over a huge range, making it a poor proxy for sharp concentration. The shell constraint (Eq. 3.6) with closed-form projection (Eq. 3.8) is simple, cheap, and theoretically grounded. The $\varepsilon = 0.025$ choice provides appropriate slack.

**4. Breadth of experimental coverage.**
The experiments cover four tasks $\times$ three datasets for image restoration, plus three distinct scientific IPs from InverseBench with genuinely different forward models (linear scattering, Fourier subsampling, nonlinear VLBI). The scientific IP evaluation (Sec. 4.2) is particularly valuable and well-motivated — the argument that domain-specific models are often unavailable in scientific domains is compelling.

**5. Practical few-shot extension.**
The few-shot variant (Eq. 3.9) with learned simplex weights over instance exemplars is a natural and well-designed extension. The softmax reparameterization handles the scale ambiguity between $\alpha_t$ and $w$ cleanly, and the implicit sparsity promotion via the simplex constraint is a nice touch.


## Weaknesses

**1. Limited novelty of individual components.**
Both core components are relatively straightforward modifications of the plug-in optimization (Eq. 2.3). The warm-start amounts to optimizing over $(z, t)$ instead of $z$ alone; the Gaussianity constraint is a projected gradient step onto a shell. The authors acknowledge (Sec. 3.2) that spherical constraints for diffusion/FM latents are not new (Yang et al., 2024). The contribution is more in the careful combination and the diagnostic analysis than in algorithmic novelty. This is not disqualifying, but the paper should frame its contributions more precisely.

**2. The simple-distortion assumption is restrictive and not always clearly satisfied.**
The warm-start strategy assumes $x \approx y$, which holds for mild degradations but breaks down for severe IPs. For 4× super-resolution ($128 \to 512$), $y$ and $x$ have different dimensions, so the closeness assumption requires interpolation (not discussed). For heavy Gaussian blur ($\sigma = 3.0$, kernel size 61), the distortion is substantial. The paper does not quantify or discuss the regime where the approximation $z_t \approx \alpha_t y + \beta_t z$ (Eq. 3.4) breaks down, or how the learned $t$ adapts. Showing the distribution of learned $t^*$ values across tasks and difficulty levels would be very informative.

**3. FMPlug still substantially trails domain-specific priors on MRI.**
Tab. 4 shows a large gap between FMPlug (23.60 dB) and Red-Diff with domain-specific priors (28.71 dB) on MRI — a gap of over 5 dB in PSNR. The authors attribute this to unrepresentative few-shot guidance (Fig. 8), which is a reasonable hypothesis, but it also reveals a fragility: the method's performance is highly sensitive to the quality and representativeness of the few-shot instances. No analysis quantifies this sensitivity (e.g., performance vs. number of instances, or vs. distance between guidance and test samples).

**4. Computational cost is high.**
Tab. 6 shows FMPlug takes 492s per image — comparable to D-Flow (456s) and roughly 50× slower than FlowDPS (10s) and 70× slower than FlowChef (7s). While the quality improvements are real, the practical impact is limited if each image takes ~8 minutes. The paper mentions that LBFGS is incompatible with multiple parameter groups as the reason for not using it, but does not explore other second-order or quasi-Newton alternatives that could accelerate convergence.

**5. Ablation is incomplete.**
The ablation (Tab. 5) only covers Gaussian deblurring on DIV2K. Given that the warm-start relies on $x \approx y$ and the scientific IPs violate this (using few-shot instances instead), ablating on scientific IPs would be important to understand which component matters in each regime. Additionally, there is no sensitivity analysis for $\varepsilon$ (the shell thickness), NFE count, or the number of optimization iterations $N$.

**6. No-reference metrics consistently favor baselines.**
FMPlug systematically underperforms on CLIPIQA and MUSIQ across almost all tasks and datasets (Tab. 3, appendix tables). While the authors argue these are secondary metrics (App. A.5), this pattern suggests that the method may sacrifice perceptual sharpness and naturalness for pixel-level fidelity — a well-known tradeoff in image restoration. This deserves more discussion than the current dismissive treatment.

**7. Non-convexity of the optimization is not discussed.**
The optimization in Eq. 3.7 is highly non-convex (the generator $\mathcal{G}_\theta$ is a deep network, and the loss landscape over $(z, t)$ has many local minima). The paper does not discuss initialization sensitivity, convergence behavior, or whether multiple restarts are needed. The image regression experiment (Tab. 2) shows that even with $\mathcal{A} = \text{Id}$, FMPlug achieves only 37.9 dB — indicating that the optimization landscape is challenging even in the simplest case.

---

> ### Author Rebuttal · Authors · 2026-03-31
>
> We thank the reviewer for their insightful comments\! We address the questions and concerns below; we will update our paper to reflect these points later.
>
> **W1: Novelty**
>
> + While FMPlug's components are simple, joint optimization over $(\\mathbf{z},t)$ is novel; the spherical constraint is not novel, as we acknowledge in our writing.
> + Identifying the fundamental deficiency of foundation (FD) FM models for solving IPs is also a major novelty of our work. Prior work on FD FM models never compares with DS and untrained priors (as we do in Tab 1 and Fig 2), suggesting they are ignorant of the issue.
> + Our method pioneers the application of FD FM models to out-of-domain scientific IPs, another novelty of our work.
>
> **W2 & Q1, Q5: Simple-distortion assumption is restrictive**
>
> + We agree that the simple-distortion assumption may look restrictive, but it holds for most image restoration problems studied in the literature. When it indeed fails, the few-shot setting is a fallback. We do not claim to solve all IPs, especially with FD FM priors.
> + Please check **Re: W1 & Q2: Dimension mismatch in $\\mathbf x$ and $\\mathbf y$** for **reviewer V4tS** above.
> + As suggested, we evaluate $t^\*$ across Gaussian deblurring degradation levels (radius $\\sigma$). As expected, higher degradation yields a smaller $t^\*$ (closer to the source). We will scale up this study during our revision.
>
> | Radius | PSNR↑ | $t^\*$ |
> | :--- | :---: | :---: |
> | 1 | 32.24 | .83 |
> | 3 | 26.58 | .78 |
> | 6 | 24.13 | .72 |
>
> **W3 & Q6: FMPlug trails DS priors on MRI**
>
> + Yes, our few-shot setting relies on the availability of **truly close** instances to the target. When this is violated, performance may suffer.
> + Our goal is to bridge the gap between FD and DS priors, though completely closing it may be impossible given their different natures.
> + Ablating the number of few-shot instances ($k$) on MRI (6 total instances) shows monotonic scaling ($k=1$ already outperforming $k=0$):
>
> | $k$ | PSNR↑ | SSIM ↑ | data misfit↓ |
> | :--- | :---: | :---: | :---: |
> | 0 | 16.36 | .39 | 37.66 |
> | 1 | 22.22 | .48 | 34.17 |
> | 3 | 22.36 | .49 | 33.97 |
> | Full |  **23.60** | **.52** |  **33.18** |
>
> **W4: Computational cost**
>
> + We acknowledge our much higher computational cost. In this paper, we prioritize recovery quality over inference speed, as we believe the qualities achieved by current methods are still far from being practically useful.
> + We agree that exploring more efficient solvers is an important future direction, currently limited by the lack of efficient implementations of second-order or quasi-Newton solvers that can reliably solve deep learning problems.
>
>
> **W5 & Q2: Limited ablation**
>
> + We perform sensitivity analysis on the shell thickness and NFE.
>
> | Thickness | PSNR↑ | SSIM ↑ | LPIPS ↓ |
> | :--- | :---: | :---: | :---: |
> | 0 | 26.45 | .74 | .41 |
> | .025 | **26.58** | .74 | .41 |
> | .1 | 26.44 | .74  | .41 |
>
> The thickness results above are consistent with the analysis in our Fig 4: $2.5\\%$ covers most of the high-density region, so it performs better.
>
> | NFE | PSNR↑ | SSIM ↑ | LPIPS ↓ |
> | :--- | :---: | :---: | :---: |
> | 1 | 26.44 | .74 | .41 |
> | 3 | 26.58 | .74 | .41 |
> | 5 | 26.62 | .74 | .41 |
>
> Our method does not require a large NFE to achieve high performance as the warm-start intrinsically reduces ODE solver approximation errors during generation. Thus, it remains robust with low NFEs.
>
> + We use adaptive early stopping, so solvers typically halt before reaching the maxiter.
> + Due to time constraints, we defer other suggested ablations to our revision.
>
> **W6: Full reference metric**
>
> + We agree the discrepancy deserves careful discussion; we will do it in our revision.
>
> **W7 & Q4: Optimization**
>
> + The imperfect (already excellent) regression result could be due to the solver's low numerical precision, the model's capacity, and the challenging landscape, among other factors.
> + To validate this, we perform a regression only using LoRA to finetune the decoder. We obtain 43dB PSNR, suggesting that the decoder's capacity is crucial.
> + Gaussian deblurring (10 random initializations per image) shows very low variance, making multiple restarts unnecessary:
>   Method | PSNR↑ | SSIM ↑ | LPIPS ↓ |
>   | :--- | :---: | :---: | :---: |
>   | FlowDPS | 20.65(.163) | .47(.11) | .58(.017) |
>   | FlowChef | 20.77(.047) | .49(.001) | .62(.002) |
>   | D-Flow | 24.43(.625) | .65(.025) | .50(.028) |
>   | **Ours** | **26.60(.025)** | **.74(.001)** | **.40(.004)** |
>
> **Q3: Few-shot weights**
> We analyze the correlation of learned weights and instance-truth $\\ell\_2$ distance over 96 test instances. We find a strong negative correlation: Spearman (-0.71) and Pearson (-0.79), ensuring closer instances correctly receive larger weights.
>
> **Limitations**
> We will follow the suggestions to expand our discussion of the limitations in our revision.

---

> > ### Author Rebuttal · Reviewer_1Nvs · 2026-04-04
> >
> > The rebuttal has addressed my concerns satisfactorily across the board, and I am comfortable maintaining my Accept recommendation.
> >
> > On the simple-distortion assumption (W2, Q1, Q5): the table showing that the learned  decreases monotonically with degradation severity is exactly the empirical validation I asked for, and directly confirms the theoretical prediction of Sec. 3.1. The clarification that the few-shot setting serves as a fallback when the assumption fails is a reasonable and honest scoping of the method.
> >
> > On the MRI gap (W3, Q6): the ablation over the number of few-shot instances showing monotonic improvement with  is informative, and the acknowledgement that completely closing the gap with domain-specific priors may be impossible given their different natures is an honest and appropriate framing.
> >
> > On the shell thickness sensitivity (W5, Q2): the ablation confirms robustness around , consistent with the theoretical analysis in Fig. 4. The NFE robustness result is also reassuring and well-explained by the warm-start reducing ODE approximation error.
> > On optimization variance (W7, Q4): the 10-restart experiment showing very low variance for FMPlug (PSNR std of 0.025) compared to D-Flow (0.625) is a strong and direct answer. The LoRA decoder experiment attributing the regression gap to capacity rather than landscape pathology is also convincing.
> >
> > On the few-shot weights (Q3): the strong negative correlation between learned weights and instance-truth distance (Spearman , Pearson ) directly confirms that the method behaves as intended.
> > On no-reference metrics (W6) and limitations: the commitment to expand discussion in the revision is appropriate.
> > The one remaining practical concern — computational cost (~8 min/image) — is honestly acknowledged, and the authors’ position that quality is the current priority is a defensible stance given the state of the field. I am satisfied with the rebuttal overall.​​​​​​​​​​​​​​​​

---

> > > ### Author Response · Authors · 2026-04-07
> > >
> > > Thank you for your thorough and encouraging review. We are thrilled that you found our mathematical grounding elegant and our experimental evaluations comprehensive. We also appreciate your follow-up acknowledgment regarding the ablation studies we provided. Thank you for your time, your constructive questions, and your strong support of our paper.

---

### Official Review · Reviewer_zY5u · 2026-03-12

**Soundness:** 2
**Presentation:** 2
**Significance:** 3
**Originality:** 2
**Overall Recommendation:** 4
**Confidence:** 4

**Summary:**

This paper focuses on solving inverse problems using pretrained flow-matching generative models. The authors note and empirically demonstrate that domain-agnostic "foundation" flow-matching priors often perform worse than their "domain-specific" counterparts. To address this, they focus on a plug-in approach that aims to estimate the optimal input to the flow-matching procedure. This optimal input is defined as a time-dependent weighted sum of the measurement and noise. They propose to learn the noise vector and the time step jointly, applying an additional sharp Gaussian regularization to the noise. Additionally, they provide a refined, few-shot version of their method tailored for scientific inverse problems, utilizing a small set of similar reference images. Finally, they validate their approach through numerical experiments, demonstrating improved performance compared to a select baseline of recent flow-matching inverse solvers.

**Compliance With Llm Reviewing Policy:**

Affirmed.

**Final Justification:**

My final recommendation is to weakly accept. The authors seem to succeed empirically in achieving better results with the use of foundation models than previous methods; however, I still believe some concepts need to be clarified in the paper, e.g., the role of training with more data should be separated from the operation of the models in the pixel versus latent space.

**Key Questions For Authors:**

1. There are two distinct concepts: (i) pixel-space versus latent-space flow-matching models, and (ii) models trained on domain-specific data versus general data. Foundation models for flow-matching are typically latent-space models trained on general data. Could you specify this distinction more clearly in the text? For example, in Table 1, do all the compared models operate in the latent space (meaning the only difference is the training data), or are some of them pixel-space models? It would also be beneficial to explicitly benchmark pixel-space versus latent-space models to isolate this factor.

2. Please specify the dimensionality of the vectors you use more clearly. For example, in Section 3.1 under the "Our time-dependent warm-up strategy" paragraph, you assume that $x$ is close to $y$. How do you deal with this assumption in the case of image super-resolution? In that task, the spatial dimensions of the measurement are clearly different from those of the ground truth image. The answer is likely that you are operating in the latent space after some preprocessing, but this dimensional matching needs to be made explicit in the text.

3. In the few-shot scientific example, how do you have access to similar images in practice?

**Limitations:**

Yes

**Strengths And Weaknesses:**

While the use of foundation models to solve inverse problems is an interesting and valuable research direction, this paper has several presentation issues that should be addressed (detailed in the questions below). Additionally, the proposed solution relies heavily on empirical results and lacks strong theoretical justification. Furthermore, the current title reads as an overstatement of the actual contributions, and the manuscript notably lacks a formal conclusion section. The abstract could also be improved for clarity; for example, introducing the "sharp Gaussian constraint" without prior context is confusing, even though the concept becomes clear later in the main text. Finally, there are duplicate entries in the bibliography that need to be corrected.

---

> ### Author Rebuttal · Authors · 2026-03-30
>
> We thank the reviewer for their insightful comments\! We address the questions and concerns below; we will update our paper to reflect these points later.
>
> **Re: W: Empirical results without strong theoretical justification**
>
> + We agree that the superiority of our method is largely demonstrated by empirical results, even though **we have provided theoretical arguments to explain why D-Flow is problematic and to motivate our algorithm (see Secs 3.1 & 3.2**). We also agree that strong methodological work should ideally be supported by a solid theoretical justification. But we strongly believe that ICML is open to anything across the whole spectrum of theoretical and empirical work, and many strong empirical papers have been accepted by ICML in the past—which have often motivated theoretical studies later.
> + Specific to solving IP with pretrained flow-matching (or diffusion) priors, most published papers in this domain at top ML conferences consist solely of empirical results, similar to our style. We suspect that such theoretical building is likely difficult, if not impossible, without strong (if not unrealistic) assumptions about the pretrained FM priors, given the highly nonlinear nature of the prior and, perhaps, the forward model as well.
>
> **Re: W: Title overstatement of actual contribution**
>
> + If the concern is the word "saving", we use it simply to highlight that FD models currently underperform domain-specific (DS) and untrained priors, and our method rescues them. We welcome suggestions for a more rigorous title and ask the reviewer to clarify which part is overstated.
>
> **Re: Writing (Conclusion, abstract clarity, duplicated citations)**
> Thank you for your careful reading\! We will revise the abstract to improve clarity, remove duplicated D-Flow citations, and add a formal conclusion section if space allows.
>
> **Re: Q1: Latent vs. Pixel space and FD vs. DS models**
>
> + We will make the distinction between latent/pixel space & FD/DS models more explicit in our revision, perhaps by color-coding in tables and highlighting in the text.
> + **The focus of the paper is on FD models trained on general images**, as we highlight in the title and writing. Most (if not all) commonly used publicly available FD FM models for general images are **latent models**. It is true that the DS models we experiment with are pixel-space models, and the difference between the latent and pixel spaces might contribute to the performance gap. Our goal is to use **the most commonly used** pretrained DS models for IP solving—which happen to be pixel-space models—to provide a performance reference to show the performance gap.
> + Even if this performance gap can change if we switch from a pixel-space DS model to a latent-space DS model, the gap between the latent-space FD model and DIP—which is an untrained prior, so no need to distinguish latent vs. pixel spaces—remains true.
> + Our overall goal is not to outperform any DS priors, which may or may not be possible, but to improve the performance when working with latent-space FD models as much as possible to reduce the gap.
>
> **Re: Q2: Be explicit about dimension mismatch in $\\mathbf x$ and $\\mathbf y$**
> Thank you for bringing this out\! We will carefully address this in our revision:
>
> + For the simple distortion setting, we assume that $\\mathbf x$ and $\\mathbf y$ (or a simple transformation of $\\mathbf y$ ) are close. So, whenever possible, we perform a simple transformation on $\\mathbf y$ to match $\\mathrm{dim}(\\mathbf x)$. For example, for super-resolution, we upsample $\\mathbf y$; for compressed sensing and all linear inverse problems with $\\mathrm{dim}(\\mathbf y) \\ne \\mathrm{dim}(\\mathbf x)$, we take $\\mathbf A^\\dagger \\mathbf y$, where $\\mathbf A^\\dagger$ is the pseudoinverse of the linear operator $\\mathbf A$.
> + When such simple transformations are not possible, generally, our simple distortion assumption is violated, and we recommend transitioning to our few-shot setting. We will include a clear step-by-step pipeline in the revision.
>
> **Re: Q3: Access similar images in the few-shot scientific setting**
>
> + For numerous scientific domains, strong knowledge about the objects to be recovered is available ahead of time, e.g., MRI scan of the brain, microscopical imaging of certain nanomaterials or biomaterials, etc. So, the recovery is in a “narrow-domain” scenario, where previous samples from similar domains can serve as the few-shot instances.
> + It is possible that only one or two of the examples out of the several provided examples are close to the true $\\mathbf x$. We do not know this in advance, but leave it to our algorithm to perform automatic selection via weight assignment.
> + Moreover, in Sec 4.2, we show that even when the provided few-shot instances differ significantly in visual appearance from the target, our proposed method still successfully leverages their shared domain structure to produce reasonable reconstructions.

---

> > ### Author Rebuttal · Reviewer_zY5u · 2026-04-03
> >
> > Thanks. I think a title that says "improving" instead of "saving" would be better, e.g., improving the solution of inverse problems with foundation models. In light of your responses, I would like to increase my score to weak accept.

---

> > > ### Author Response · Authors · 2026-04-07
> > >
> > > Thank you for your thoughtful review and for your active engagement during the rebuttal phase. Thank you as well for raising your score in light of our clarifications. Your feedback has significantly strengthened the presentation of our work.

---

### Official Review · Reviewer_V4tS · 2026-03-13

**Soundness:** 3
**Presentation:** 3
**Significance:** 2
**Originality:** 2
**Overall Recommendation:** 4
**Confidence:** 4

**Summary:**

This paper introduces FMplug, a framework that leverages flow matching to solve inverse problems by combining instance-guided, time-dependent warm-start strategies with Gaussian regularization. The approach positions itself as an improvement over prior plugin-style and D-flow methods, addressing specific limitations in their optimization dynamics.

**Compliance With Llm Reviewing Policy:**

Affirmed.

**Final Justification:**

The reply answered my question, so I'm inclined to keep the score.

**Key Questions For Authors:**

1. From Algorithm 1, it appears that the method in this paper employs an adaptive selection algorithm for the time t of the plugin/d-flow (using gradient updates). I would like to know if my understanding is correct, or could the authors clarify the distinction?

2. Algorithm 1 initializes the noise variable to match the dimension of y. How does the method handle cases where dim(y) ≠ dim(x)? Is there an implicit upsampling step, or does the algorithm assume equal dimensions? What happens empirically when this gap is large?
﻿
3. For the scientific reasoning experiments, which pre-trained flow matching model was used?

**Limitations:**

yes

**Strengths And Weaknesses:**

**Strengths**

1.  The paper provides a detailed critique of PLugin and D-flow, identifying their failure modes and using these as concrete motivations for the proposed method. This grounding in related literature strengthens the paper's narrative.
﻿
2. The inclusion of scientific reasoning experiments and a few-shot variant of the algorithm demonstrates awareness of practical deployment constraints and broadens the paper's applicability beyond standard image inverse problems.
﻿
3. The instance-guided, time-dependent initialization is a sensible heuristic that is easy to understand and likely contributes to faster convergence in practice. The authors explain its motivation clearly.

**Weaknesses**

1. Dimension compatibility concern in Algorithm 1. As shown in line 4 of Algorithm 1, the method directly operates on a noise variable initialized to match the dimension of y. However, in many practical inverse problems (e.g., super-resolution, compressed sensing), the observation y lives in a lower-dimensional space than the signal x. It is unclear how the algorithm handles this mismatch. Does the method require a projection or upsampling step? If the algorithm silently assumes dim(y) = dim(x), this represents a significant limitation that is not acknowledged. The authors should clarify this and, if a dimension gap exists.
﻿
2. The authors assert that FMplug can be applied to scientific inverse problems. However, many scientific domains (e.g., data assimilation, MRI reconstruction) operate in non-RGB, non-image spaces with complex forward operators. The flow matching prior trained on natural image distributions may not transfer meaningfully to these settings. The paper does not address how the method would be adapted when a suitable pre-trained flow matching model does not exist for the target domain. The experiments in the paper appear to remain within or close to the RGB image regime, which makes the claim of general scientific applicability overstated.
﻿
3.  While the paper frames FMplug as a unified framework, the core technical contributions — gradient-based time selection and Gaussian regularization — are incremental modifications to existing plugin-style inference methods.

---

> ### Author Rebuttal · Authors · 2026-03-30
>
> We thank the reviewer for their insightful comments\! We address the questions and concerns below; we will update our paper to reflect these points later.
>
> **Re: W1 & Q2: Dimension mismatch in $\\mathbf x$ and $\\mathbf y$**
> Thank you for pinpointing this omission\!
>
> + For the simple distortion setting, we assume that $\\mathbf x$ and $\\mathbf y$ (or a simple transformation of $\\mathbf y$ ) are close. So, whenever possible, we perform a simple transformation on $\\mathbf y$ to match $\\mathrm{dim}(\\mathbf x)$. For example, for super-resolution, we upsample $\\mathbf y$; for compressed sensing and all linear inverse problems with $\\mathrm{dim}(\\mathbf y) \\ne \\mathrm{dim}(\\mathbf x)$, we take $\\mathbf A^\\dagger \\mathbf y$, where $\\mathbf A^\\dagger$ is the pseudoinverse of the linear operator $\\mathbf A$.
> + When such simple transformations are not possible, generally, our simple distortion assumption is violated, and we recommend transitioning to our few-shot setting. We will include a clear step-by-step pipeline in the revision.
>
> **Re: W2: Image modality mismatch in scientific inverse problems (IPs)**
>
> + We would like to direct the reviewer to Section 4.2, where we provide experiments on scientific IPs. Specifically, linear inverse scattering and black hole imaging recover single-channel images, whereas MRI recovers complex-valued images. Although we omitted the implementation details regarding channel mismatch in the main text due to space constraints, the exact procedures are documented in Appendix A.8. We will add a clear pointer in the main text.
> + For general scientific IPs, we believe that so long as the recovery target is image-like spatial data, single or multiple channels, one can reasonably modify the RGB priors in foundation FM models to help. However, we also agree that there are scientific domains where the recovery target is not image-like spatial data at all, for which foundation FM models might not be helpful.
>
> **Re: W3: FMPlug as incremental modifications to existing plugin-style methods**
> We agree that FMPlug builds on existing plug-in frameworks. Below, we would like to reiterate the other important contributions we make in the paper beyond FMPlug.
>
> + **Identifying & addressing the foundation model gap:** While the related literature has extensively studied pretrained Domain-Specific (DS) models for solving IPs, such research using Foundation (FD) models has only begun recently. Clearly, prior work has largely overlooked the fundamental difficulty of solving IPs with FD models, as they **never** explicitly compare with DS and untrained priors (as we do in Table 1 and Figure 2\) or point out the fundamental deficiencies of FD models. **It is daunting that the pretrained FM priors even underperform the untrained priors, and on Gaussian deblurring, the recovery quality is only marginally better than the blurry image itself**. We are **the first** to pinpoint the gap and take the **first steps** to address it. Honestly, we believe identifying the gap for the community is perhaps even more important than our first step toward addressing it.
> + **Out-of-domain scientific IPs:** We successfully extend the use of pretrained FD models to scientific IPs. We believe we are the first to push FD models to achieve strong performance in out-of-domain scientific IPs, especially in scenarios where training DS models is infeasible due to data or resource limitations.
>
> **Re: Q1: Clarification on learnable $t$**
> Yes, $t$ is a learnable parameter adaptively learned alongside the seed $\\mathbf z$ using the typical gradient descent methods, instead of the fixed $t$ used in D-Flow. This is a crucial ingredient of our algorithm.
>
> **Re: Q3: FM model for scientific IPs**
> We **exclusively** use SD3 as our base model **for all tasks**. This reliance on a single, unified foundation model underscores the generality of our approach and fundamentally distinguishes our work from existing literature.

---

> > ### Author Rebuttal · Reviewer_V4tS · 2026-04-04
> >
> > Thank you to the author for responding to my clarification; I would like to keep my score.

---

> > > ### Author Response · Authors · 2026-04-07
> > >
> > > Thank you for your time and for engaging with our rebuttal. We greatly appreciate your constructive feedback. Your insights have helped us clarify the practical scope and implementation details of our method. We are grateful for your review and your recommendation.

---

### Official Review · Reviewer_Yt6D · 2026-03-15

**Soundness:** 3
**Presentation:** 2
**Significance:** 2
**Originality:** 3
**Overall Recommendation:** 3
**Confidence:** 3

**Summary:**

This paper studies how to use pretrained foundation flow-matching (FM) models as priors for inverse problems.

The authors observe that existing foundation FM priors often underperform domain-specific FM priors and even untrained methods such as DIP. To address this, they propose FMPlug, a plug-in solver that improves optimization over a pretrained FM generator in two ways.

First, it introduces a learnable time-dependent warm start, jointly optimizing the latent variable and the starting point along the FM trajectory. Second, it enforces a thin-shell constraint that keeps the latent close to the Gaussian source geometry used during training.

The approach is further extended to few-shot scientific imaging, where the warm start is replaced with a simplex-weighted combination of a few guidance images. Experiments cover multiple natural-image restoration tasks and three scientific inverse problems (LIS, MRI, and black-hole imaging).

Results show consistent improvements over prior foundation-FM solvers and competitive performance with DIP in several settings.

**Compliance With Llm Reviewing Policy:**

Affirmed.

**Key Questions For Authors:**

* What is the exact optimization objective used in FMPlug?  Eq. (2.3) includes a prior term $(\Omega\circ G_\theta)$, but the FMPlug algorithms appear to optimize only the data term with the shell constraint. Is $\Omega$ removed or absorbed into the constraint?

* How is the warm start implemented for latent FMs such as SD3?

* Can the sparsity claim for the few-shot simplex weights be justified or revised? As The simplex constraint alone does not induce sparsity in the usual L1 sense since all simplex points share the same L1 norm.

* How were the baselines tuned? Providing matched-compute or matched-budget comparisons would help ensure fairness.

**Limitations:**

Yes

**Strengths And Weaknesses:**

# Strengths
The paper addresses a timely and practically relevant problem: how to leverage widely available foundation generative models for inverse problems where domain-specific priors are unavailable. The proposed diagnosis, i.e. existing FM plug-in methods may initialize optimization in regions misaligned with the Gaussian source geometry, is interesting, and the resulting solver design (joint optimization over latent and time with a thin-shell constraint) is simple and well motivated.

# Weaknesses
The main concern is technical justification. The thin-shell argument is motivated by concentration-of-measure intuition but remains heuristic, and the method only constrains the latent radius rather than preserving the full Gaussian distribution. The paper also does not convincingly demonstrate that off-shell initialization is the dominant failure mode for FM-based inverse solvers.

There are also several technical clarity issues. The plug-in objective in Eq. (2.3) includes a prior term $(\Omega\circ G_\theta)$, but the FMPlug algorithms appear to optimize only the data term with the shell constraint. The latent-space implementation for SD3 is not fully specified, and some details in the algorithms (e.g., enforcing $t\in [0,1]$ or gradient notation in Alg. 2) are unclear.

In experimental parts,  while the empirical results are promising, the baseline tuning and evaluation protocol are not fully described, and results are reported without variance across seeds or initializations. This makes it difficult to assess the robustness of some gains.

---

> ### Author Rebuttal · Authors · 2026-03-31
>
> We thank the reviewer for their insightful comments\! We address the questions and concerns below; we will update our paper to reflect these points later.
>
> **Gaussianity not preserved**
> Ideally, we would specify $\\mathbf{z} \\sim \\mathcal N(\\mathbf 0, \\mathbf I)$ as the constraint to strictly enforce Gaussianity. But we are not aware of any computationally feasible way of dealing with this **stochastic constraint** directly. Natural ideas, such as using distributional-distance regularization (e.g., KL, Wasserstein) and log-likelihood regularization (e.g., on $\\| \\mathbf z \\|\_2^2$ as done in D-Flow or on $\\mathbf z$ itself), including our concentration-shell regularization, are all **relaxations**. However, they differ in faithfulness to Gaussianity; hence, we highlight the looseness of the D-Flow’s log-likelihood regularization and favor our concentration-shell regularization.
>
> **Off-shell initialization as dominant failure mode**
> Thank you for bringing this out\! We augment our theoretical argument in the 1st paragraph of Sec 3.1 with the following numerical results.
>
> + Replacing D-Flow’s initialization with our warm-start improves performance:
>
> | Method | PSNR↑ | SSIM ↑ | LPIPS ↓ |
> | :--- | :---: | :---: | :---: |
> | Org | 24.43 | 0.65 | 0.50 |
> | Ours | **25.93** | **0.71** | **0.42** |
>
> **Regularization term $\\Omega$ missing for FMPlug**
> Eq. (2.3) is a **generic** formulation for all inverse problems (IPs); for particular IPs, regularization $\\Omega$ may or may not be needed. For FMPlug, the constraint on $\\mathbf z$ can be viewed as a regularization for the following reason: if we define the shell constraint set as $S$, then the constraint can be written as $\\mathbf z \\in S$. Now introduce a set-indicator function $\\delta(z)$ defined as
> $$\\delta\_S(\\mathbf z) \= \\begin{cases} 0 & \\mathbf z \\in S \\\\ \\infty & \\mathbf z \\notin S. \\end{cases}$$
> Then one can equivalently replace the constraint by $\\delta\_S(\\mathbf z)$ as a regularizer. In other words, the set-indicator conversion unifies constrained formulations into regularized formulations. So, for FMPlug, the constraint is the regularization itself.
>
> **Warm start for latent FMs and other implementation details**
>
> + For latent FM warm starts, we simply replace $\\mathbf{y}$ with $\\mathcal{E}\\circ\\mathbf{y}$, and $\\mathcal{A}\\circ\\mathcal{G}\_\\theta$ with $\\mathcal{A}\\circ\\mathcal{D}\\circ\\mathcal{G}\_\\theta$ in Eqs. (3.5) and (3.7) (briefly mentioned around Line 265).
> + $t$ is reparametrized as $\\mathrm{sigmoid}(s)$ and hence eliminated from the constraint.
>
> We will make all such missing details explicit and sharp in our revision.
>
> **Sparsity claim in the few-shot simplex constraint**
> We agree the current phrasing lacks rigor and will revise it. Our goal is to encourage sparsity so the model filters out irrelevant few-shot instances. In future work, we plan to explore explicit sparsity regularization, e.g., using $|\\mathbf{z}|\_2=1$ to fix scaling and $|\\mathbf{z}|\_1$ as the regularizer.
>
> **parameter search and robustness of the results**
>
> + For FlowChef and FlowDPS, we use the default hyperparameters reported in their papers. This is because, for experiments involving FlowChef and FlowDPS, we follow their experimental settings. It is reasonable to assume their authors have conducted sufficient hyperparameter tuning to optimize performance.
> + For D-Flow and our method, we use \[Optuna\](https://optuna.org/), a state-of-the-art hyperparameter search package,  to conduct hyperparameter search with a limited budget (\`n\_trials=20\`). We round the resulting hyperparameter values for simplicity.
> + To study the robustness of the experimental results, during the rebuttal period, we conduct a quick experiment on a subset of the DIV2K dataset for Gaussian deblurring (20 images, 10 random initializations each). The table below shows our method **consistently** and **robustly** outperforms the baselines. We plan to include the standard deviation for all our experimental results.
>
> | Method | PSNR↑ | SSIM ↑ | LPIPS ↓ |
> | :--- | :---: | :---: | :---: |
> | FlowDPS | 20.65(0.1632) | 0.47(0.011) | 0.58(0.017) |
> | FlowChef | 20.77(0.047) | 0.49(0.001) | 0.62(0.002) |
> | D-Flow | 24.43(0.625) | 0.65(0.025) | 0.50(0.028) |
> | **Ours** | **26.60(0.025)** | **0.74(0.001)** | **0.40(0.004)** |
>
> **Fair evaluation protocol**
> While practical algorithms must balance output quality and computational cost, current methods (including ours) still fall short of satisfactory visual results for these tasks (e.g., PSNRs generally remain below the practical threshold of 30). Therefore, we prioritize output quality in this work. Once quality reaches an acceptable practical baseline, optimizing computational cost becomes the next logical step.

---

> > ### Author Rebuttal · Reviewer_Yt6D · 2026-04-05
> >
> > Thank you for the thoughtful rebuttal. I appreciate the clarifications on the shell constraint as a regularizer, the added implementation details and the acknowledgement of the simplex sparsity issue.
> >
> > However, my main concerns remain. The core justification is still largely heuristic (especially the thin-shell argument), key implementation details are not yet clearly reflected in the paper, and experimental fairness/robustness is only partially addressed (e.g., uneven tuning across baselines, limited variance analysis, and lack of matched-budget comparisons). The few-shot extension also remains somewhat under-motivated after revising the sparsity claim.
> >
> > Overall, while the rebuttal is helpful, it does not change my assessment. I keep my score at 3.

---

> > > ### Author Response · Authors · 2026-04-07
> > >
> > > Thank you again for acknowledging our rebuttal\! We appreciate your continued engagement and want to briefly address your final notes:
> > >
> > > **Re: The Gaussianity Regularization:** If you have any **specific suggestions** for making it rigorous, we would be very grateful to hear them and incorporate them.
> > >
> > > **Re: Updates to the Manuscript:** While it is **not allowed to modify the manuscript** during the rebuttal, we will update the promised changes, including the missing details, in the future version.
> > >
> > > **Re: Experimental Fairness** We have tried our best to ensure the hyperparameter tuning was fair by relying on the highly optimized default settings from the FlowChef and FlowDPS papers, while using *Optuna, an automatic parameter tuning package,* with a fixed budget for the others. As promised, we will add the full variance analysis into the revision to demonstrate robustness.
> > >
> > > **Re: Matched-Budget:** We respectfully disagree with prioritizing it at this stage, because the absolute performance of **all** existing FD-based methods on these tasks remains relatively poor (e.g., falling short of practical PSNR thresholds). Our primary goal in this paper is to establish a method that first achieves an acceptable quality; optimizing the computational budget will be the immediate next step.
> > >
> > > **Re: Few-Shot Weights:** Regarding the validity of the few-shot formulation, we agree that the simplex constraint does not necessarily induce sparsity, but reasonable combinations of the instances do not have to be sparse either. The key is that closer instances get larger weights and remote instances get smaller weights. This is confirmed by our empirical results—the correlation between the learned weights and the instance-target distances is strongly negative: Spearman's (-0.71) and Pearson's (-0.79), implying that closer instances receive larger weights.
> > >
> > > Thank you again for helping us to make this work stronger and clearer\!

---

### Decision · Program_Chairs · 2026-04-30

**Decision:**

Accept (regular)

**Comment:**

The reviews are mostly positive and the paper received scores of 3, 4, 4, and 5. The rebuttal was comprehensive and responsive, and Reviewer zY5u raised their score to 4, and Reviewer 1Nvs confirmed all concerns were resolved. The only dissenting voice is Reviewer Yt6D (confidence 3), who maintained a weak reject primarily on grounds of heuristic justification of thin Gaussian shell assumption. However,  this assumption has been widely used in inverse problem community using diffusion/flow models, so it is not considered as critical issue to devaluate the contribution of this work.  Only one remaining minor issue is the overstating the contribution as reflected in the title. The authors are advised to change the title to "Improving Foundation Flow-Matching...".   With that, the paper is ready for publication.